# Toolkits for detailed and high-throughput interrogation of synapses in *C. elegans*

Maryam Majeed[1†], Haejun Han[2,3†], Keren Zhang[4†], Wen Xi Cao[1], Chien-Po Liao[1], Oliver Hobert[1]*, Hang Lu[3,4]*

[1]Department of Biological Sciences, Howard Hughes Medical Institute, Columbia University, New York, United States; [2]School of Electrical and Computer Engineering, Georgia Institute of Technology, Atlanta, United States; [3]The Parker H Petit Institute of Bioengineering and Bioscience, Georgia Institute of Technology, Atlanta, United States; [4]School of Chemical and Biomolecular Engineering, Georgia Institute of Technology, Atlanta, United States

*For correspondence:
or38@columbia.edu (OH);
hang.lu@gatech.edu (HL)

†These authors contributed equally to this work

Competing interest: The authors declare that no competing interests exist.

**Abstract** Visualizing synaptic connectivity has traditionally relied on time-consuming electron microscopy-based imaging approaches. To scale the analysis of synaptic connectivity, fluorescent protein-based techniques have been established, ranging from the labeling of specific pre- or post-synaptic components of chemical or electrical synapses to transsynaptic proximity labeling technology such as GRASP and iBLINC. In this paper, we describe WormPsyQi, a generalizable image analysis pipeline that automatically quantifies synaptically localized fluorescent signals in a high-throughput and robust manner, with reduced human bias. We also present a resource of 30 transgenic strains that label chemical or electrical synapses throughout the nervous system of the nematode *Caenorhabditis elegans*, using CLA-1, RAB-3, GRASP (chemical synapses), or innexin (electrical synapse) reporters. We show that WormPsyQi captures synaptic structures in spite of substantial heterogeneity in neurite morphology, fluorescence signal, and imaging parameters. We use these toolkits to quantify multiple obvious and subtle features of synapses – such as number, size, intensity, and spatial distribution of synapses – in datasets spanning various regions of the nervous system, developmental stages, and sexes. Although the pipeline is described in the context of synapses, it may be utilized for other 'punctate' signals, such as fluorescently tagged neurotransmitter receptors and cell adhesion molecules, as well as proteins in other subcellular contexts. By overcoming constraints on time, sample size, cell morphology, and phenotypic space, this work represents a powerful resource for further analysis of synapse biology in *C. elegans*.

## eLife assessment

Studies of synaptic development and plasticity in the nematode *C. elegans* have been limited by the difficulty of rapid, accurate assessments of synaptic structure. Here, with a series of **convincing** studies, the authors introduce and validate a **valuable** computational pipeline, "WormPsyQi," that allows rapid, reproducible quantitation of fluorescent synaptic puncta while minimizing human error and bias. The authors also describe a new set of strains carrying synaptic markers. Together, these tools should provide groups studying this model system with the ability to quantitatively characterize chemical and electrical synapses, even in densely packed regions in 3D space such as the nerve ring.

## Introduction

The nematode *Caenorhabditis elegans* was the first organism for which a full nervous system connectome was established (*White et al., 1986*). Connectomes now exist for both sexes (*Cook et al.,*

*2019*; *Hall and Russell, 1991*; *White et al., 1986*), the pharynx (*Cook et al., 2020*; *Albertson and Thomson, 1976*), for different developmental stages (*Brittin et al., 2021*; *Witvliet et al., 2021*), and for the dauer diapause stage (*Yim et al., 2023*). However, due to the laboriousness of acquiring and analyzing electron microscopy (EM) images and the resulting small sample sizes, our understanding of many aspects of synaptic connectivity has remained limited. For example, we are only beginning to understand the extent of developmental and inter-individual variability in synaptic connectivity and adjacency (*Cook et al., 2023*; *Brittin et al., 2021*; *Witvliet et al., 2021*), but we remain much in the dark of how exactly external or internal factors establish, alter, and maintain the synaptic connectome. While progress in CRISPR/Cas9 genome engineering and earlier methods continue to increase the array of mutants available to a *C. elegans* geneticist, a lack of synaptic fluorescent reporters and difficulty in scoring them remain a bottleneck for large-scale mutant analyses. This is especially true in the synapse-dense regions of the *C. elegans* nervous system, such as the nerve ring.

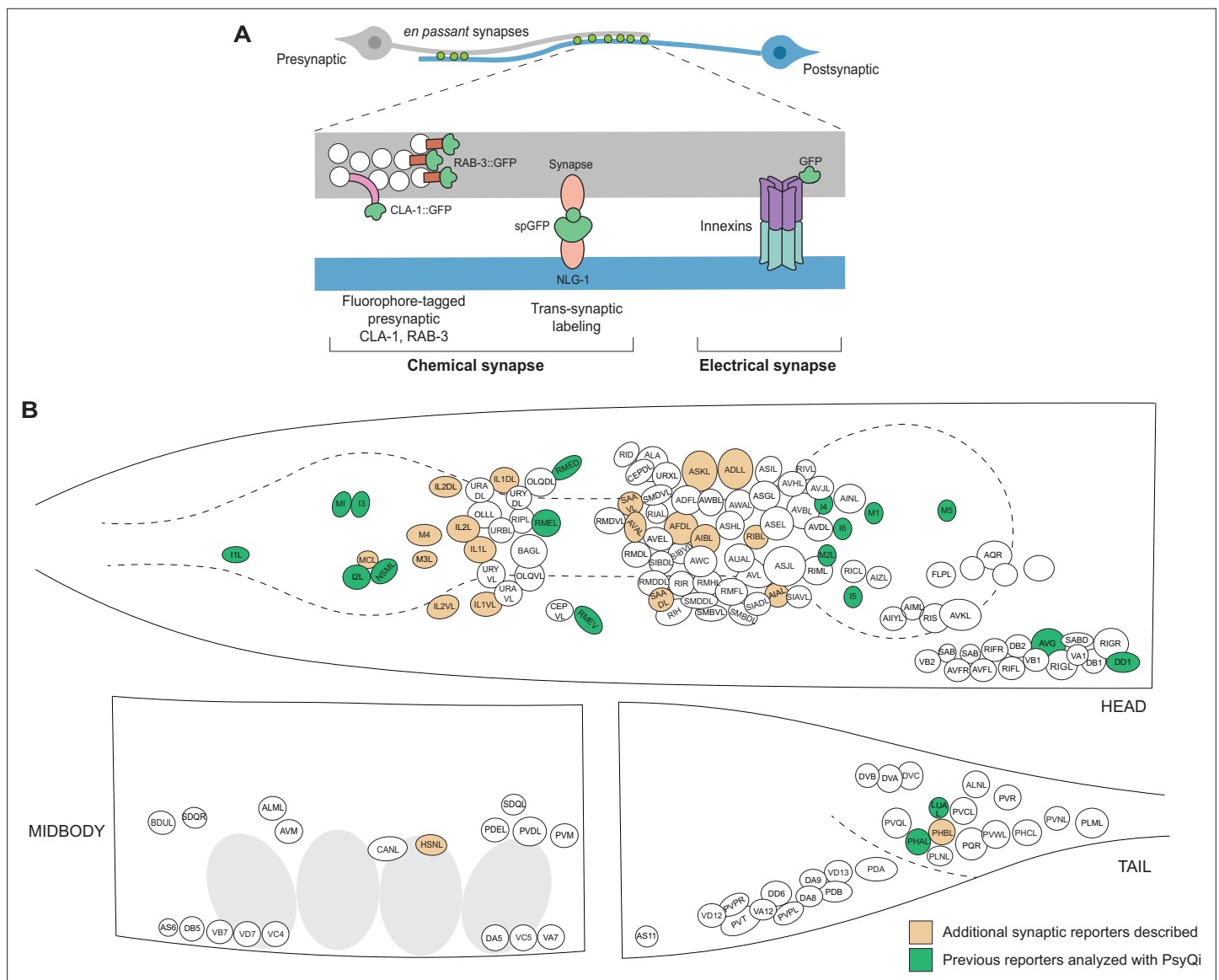

**Figure 1.** An expanded toolkit of synaptic reporters to study *C. elegans* synapses. (**A**) A schematic of reporters used to visualize *C. elegans* synapses. Chemical synapses were visualized using fluorophore-tagged CLA-1 or RAB-3 proteins to label presynaptic specializations, or synapse-specific reporters using transsynaptic tools such as NLG-1-based GRASP. Electrical synapses were visualized by directly tagging constituent innexin proteins with a fluorophore. (**B**) A schematic showing all neuron classes in the adult *C. elegans* hermaphrodite. Neurons for which new reporters are described in this paper are colored in orange. Neurons for which reporters have been previously published but are quantified with WormPsyQi in this paper are colored in green.

It has, therefore, been of paramount interest to establish alternative means to visualize synaptic connectivity that would allow us to score larger sample sizes in living animals. Classically, the study of *C. elegans* synapse biology has been aided by reporters in which various synaptic components are fluorescently labeled (*Figure 1*). For chemical synapses, these include reporters labeling pre- and post-synaptic proteins – such as SNB-1/VAMP, RAB-3, SYD-2/Liprin, CLA-1/Piccolo, and GLR-1/AMPAR – fused with a fluorescent protein (*Xuan et al., 2017*; *Mahoney et al., 2006*; *Yeh et al., 2005*; *Shen and Bargmann, 2003*; *Zhen and Jin, 1999*; *Nonet, 1999*; *Rongo et al., 1998*; *Jorgensen et al., 1995*) and split-fluorophore transsynaptic technology such as GFP Reconstitution Across Synaptic Partners (GRASP) (*Feinberg et al., 2008*) and other variants (*Feng et al., 2020*; *Feng et al., 2019*), and Biotin Labeling of Intercellular Contacts (iBLINC) (*Desbois et al., 2015*; *Figure 1A*). In the case of electrical synapses, innexins can be endogenously tagged with fluorescent proteins to visualize homo- or heteromeric electrical synapses (*Hendi et al., 2022*; *Gordon et al., 2020*; *Bhattacharya et al., 2019*; *Figure 1A*). These tools have been somewhat constrained by the relative paucity of cell-specific promoters, but recent single-cell RNA-sequencing studies (*Taylor et al., 2021*; *Packer et al., 2019*) have yielded novel cell-specific promoters for driving fluorescent proteins in neurons which previously lacked sparse-expressing genes. Moreover, the relative ease of labeling endogenous genetic loci with a fluorophore using CRISPR/Cas9 (*Eroglu et al., 2023*; *Ghanta et al., 2021*; *Ghanta and Mello, 2020*) has led to an increase in reporters with fluorophore-tagged proteins localized at chemical and electrical synapses (*Zhou et al., 2021*; *Bhattacharya et al., 2019*; *Lipton et al., 2018*; *Tu et al., 2015*; *Pinan-Lucarré et al., 2014*). With these advances over the years, the *C. elegans* community has built a growing, but still very limited, collection of synaptic reporters. These reporters have been instrumental in studying synapse biology in abundant paradigms including, but not restricted to, synaptogenesis (*Mizumoto et al., 2023*; *Kurshan and Shen, 2019*; *Jin, 2005*), sexual dimorphism (*Pechuk et al., 2022*; *Salzberg et al., 2020*; *Bayer et al., 2020*; *Bayer and Hobert, 2018*; *Cook et al., 2019*; *Hart and Hobert, 2018*; *Weinberg et al., 2018*; *Oren-Suissa et al., 2016*), and experience-dependent plasticity (*Chandra et al., 2023*; *Bhattacharya et al., 2019*; *Bayer and Hobert, 2018*; *Hart and Hobert, 2018*). A toolkit of novel reporters is presented in this paper as a resource for the *C. elegans* community (*Figure 1B*, *Supplementary file 1*).

To complement the synaptic reporters and facilitate their proper scoring, we have developed an image analysis pipeline called Worm **P**unctate **Sy**napse **Q**uantifier, or WormPsyQi, which uses machine learning to process and extract useful biological data from a large number of images in an automated, unbiased, and robust manner. By running WormPsyQi on many image datasets of novel and publicly available synaptic reporters (*Figure 1B*), we show that the pipeline can be generalized to synaptic fluorescent reporters of many types, neurons with diverse morphologies, and images taken by multiple users using different microscopes and imaging parameters.

The toolkits described in this paper (strains and imaging pipeline) mitigate several ubiquitous problems related to manual counting of puncta in large datasets. These include, but are not limited to, difficulty in scoring synapses in synapse-dense regions such as the somatic and pharyngeal nerve rings, user bias, human error, problems stemming from scoring images with low signal-to-noise ratio (SNR) or in regions with high background autofluorescence, laboriousness of scoring large datasets and puncta features such as size and intensity, and inter-dataset variability. We anticipate that the toolkits presented here will provide a motivation for the *C. elegans* research community to continue to build more synaptic reporters and to engage in an analysis of the vast number of presently unstudied synaptic connections in the *C. elegans* nervous system.

## Results and discussion
### Generating a semi-automated and robust pipeline for scoring synapses

A key advantage that fluorescent reporters provide over traditional EM-based approaches is that they allow a high-throughput and quick visualization of synapses. However, an ensuing limitation is the quantification of synapse number and ultrastructure. These include not only obvious features such as the number of synaptic puncta, but also at first indiscernible features such as relative size, fluorescence intensity, and spatial distribution of puncta along a neuron's processes.

Manual synaptic puncta quantification in large image datasets can be problematic (*San-Miguel et al., 2016*; *Crane et al., 2012*). Firstly, it is tedious and low in throughput; for neurons with high

synapse density and complex neurite morphologies in three-dimensional (3D) space, it can take up to several hours to quantify a dataset of typical size. Secondly, manual quantification requires some *a priori* information about the region of interest (ROI), thus discouraging observation outside the ROI that may be of relevance in different genetic conditions or environmental backgrounds. In addition, subtle features may escape manual quantification altogether, possibly hindering their use as a proxy for measuring neuron-specific synapse properties and impeding a comprehensive understanding of pathways that mediate precise synapse distribution. Lastly, manual quantification introduces human error and bias in cases where the fluorescent signal is dim, synapse density is high, the SNR is poor, and neurons have complex morphologies that obstruct manual counting in a 3D image stack. These problems are further compounded by dozens of labs generating various synaptic reporters and using different microscopes and imaging parameters. All these currently prevent datasets from being easily interpreted, reproduced, shared, and reused. These reasons also partially explain why we cannot rely exclusively on existing image analysis software optimized for other types of datasets or other model organisms.

To address some of these challenges, we have developed a semi-automated computational pipeline, WormPsyQi that scores synaptic puncta in *C. elegans* in a high-throughput manner and with reduced bias. The pipeline is accompanied by a Python-based graphical user interface (GUI) that allows users to seamlessly move through each step and process large volumes of data (*Figure 2—figure supplement 1B*), which greatly reduces the barrier to use. WormPsyQi consists of five key steps: neurite mask segmentation (optional), synapse segmentation, validation and correction, re-labeling and training (optional), and quantification (*Figure 2—figure supplement 1A*). First, we generate a neurite mask. Many image datasets include channels for both synaptic and cytoplasmic markers, and the latter helps exclude noise or autofluorescence signals from other tissues (e.g. the intestine). For neurite segmentation, we modified a convolutional neural network architecture, U-Net (*Guo et al., 2022*; *Guo et al., 2020*; *Ronneberger et al., 2015*), to robustly segment the neurite and soma, and create a mask that helps in specifying the ROI in which synapses are later segmented (*Figure 2—figure supplement 2A*). Masking prior to synapse segmentation is critical if the SNR is poor; it also significantly reduces the overall image processing time by constraining synapse segmentation within the neurite mask.

After masking, a local feature-based pixel classification model segments synaptic pixels inside the neurite mask (*Figure 2—figure supplement 2B*). In the absence of a mask, the pipeline will indiscriminately segment all puncta within the image. We provide both pre-trained (or default) models and an option for users to train custom synapse segmentation models. Image datasets from four diverse synaptic reporters were used to train four independent default models optimized to detect (1) small synapses (1–3 pixels radius) with low SNR (e.g. puncta with similar intensity and morphology as autofluorescence noise in surrounding tissue), (2) small synapses with high SNR, (3) medium-sized synapses (3–10 pixels radius), and (4) diffuse (>10 pixels radius) signal (see Methods for more detail). These pre-trained models can reliably cover various types of reporters, cells, and imaging parameters as discussed in later sections. To facilitate the model choice, there is an option for users to test all models on representative images, validate which model works best, and proceed with processing the entire dataset using the optimal model. In addition to this, the GUI enables users to review and edit the synapse segmentation results (*Figure 2—figure supplement 1*), as well as to train a custom model by using user-generated training data (coupled images and manual annotations of these images – for details, see Github documentation). In our experience, a minimum of one fully labeled image stack in the training set is sufficient for the segmentation accuracy to converge (*Figure 2C*).

Finally, WormPsyQi also quantifies features of predicted synapses and returns individual and summarized image- and synapse-specific features in a CSV file format, facilitating further statistical analysis. This quantification process not only yields synapse counts (*Figures 3–10*), but also provides metrics that are more challenging to discern visually. These metrics include total and average synapse volumes, spatial density of synapses, and fluorescence intensity (*Figure 10*). Such comprehensive analysis can enable in-depth analysis across different developmental stages, genetic strains, or imaging conditions. Altogether, WormPsyQi is an 'image-in-analysis-out' pipeline for quantitative analysis of synaptic puncta in fluorescent images.

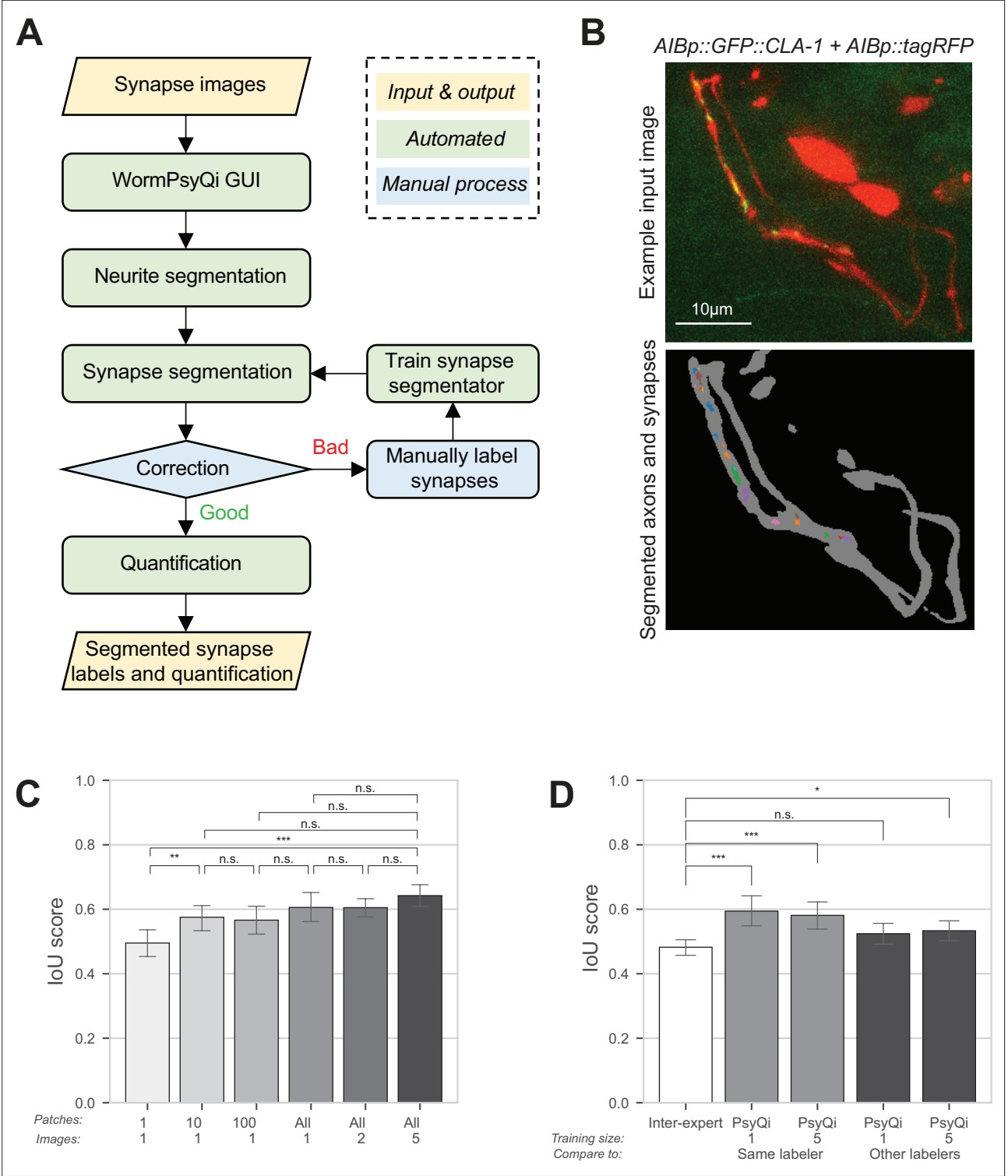

**Figure 2.** WormPsyQi robustly segments neurites and quantifies synapses. (**A**) A flowchart of the WormPsyQi pipeline. The pipeline is automated except for the correction process, which can be skipped if the initial segmentation result is deemed good enough. (**B**) Representative input raw and output segmentation images for the interneuron pair AIB. AIB-specific presynaptic specializations and processes were visualized using GFP-tagged CLA-1 (*otIs886*) and a cytoplasmic marker (*otEx8023*). Predicted puncta in the lower panel are colored arbitrarily to represent discrete fluorescent signal. Cell bodies are removed from the mask to exclude any soma-localized puncta in downstream steps. Images are maximum intensity projections of confocal Z-stacks. (**C, D**) Synapse segmentation accuracy is assessed with intersection over union (IoU) score between the ground-truth and predicted labels. Calculations were performed on six images of the I5p::GFP::CLA-1 reporter (*otEx7503*). The images were labeled by four different experts. (**C**) The IoU

*Figure 2 continued on next page*

*Figure 2 continued*

score over the size of the training set shows that having one labeled image as a training set is enough to reliably predict synapses for the remaining images in a dataset. In bars 1–3 (from left to right), one image was selected as the training set and then the given number of patches in the column label were randomly sampled after the regular patch sampling step of training (see Methods) to further reduce the size of the training set. In bars 4–6, the number of images in the column label was selected as the training set. For each expert's labels, all possible combinations of the training set were tested and the IoU score for the hold-out test set is shown. (**D**) The average IoU score between different experts' labels is taken as a benchmark (bar 1, leftmost). The IoU scores of WormPsyQi prediction are significantly greater than the benchmark IoU when the same expert's labels used in the training step were taken as the ground-truth to compare with, both when one and five images were used for training. If another expert's labels were taken as ground-truth, then the IoU score of WormPsyQi prediction was lower but still demonstrated significant improvement compared to the benchmark IoU. p values were calculated using one-way analysis of variance (ANOVA) with Bonferroni correction for multiple comparisons ***$p \leq 0.001$, **$p \leq 0.01$, *$p \leq 0.05$, and $p > 0.05$ not significant (ns).

The online version of this article includes the following figure supplement(s) for figure 2:

**Figure supplement 1.** Visual demonstration of WormPsyQi workflow and interface.

**Figure supplement 2.** Architecture overview of neurite segmentation and synapse segmentation models.

**Figure supplement 3.** The correlation between WormPsyQi and manual count is comparable to the human-to-human correlation.

**Figure supplement 4.** Comparison between WormPsyQi and human quantification for puncta features and processing time.

## WormPsyQi reliably scores synapse features and expedites analysis relative to manual scoring

To evaluate WormPsyQi's segmentation accuracy and precision, four independent experimenters (*Figure 2D*, inter-expert), manually annotated a dataset of six images, pixel-by-pixel, in which presynaptic specializations of the pharyngeal neuron I5 were labeled with GFP::CLA-1. Training models were constructed from subsets of the dataset and their performance was tested on the remaining images (*Figure 2D*). We selected the intersection over union (IoU) between expert labels and WormPsyQi's prediction compared to inter-expert label similarity to benchmark the segmentation performance. Notably, the IoU between WormPsyQi's segmentation and the original labeler's annotation was significantly higher than the inter-expert IoU (*Figure 2D*, columns 2 and 3), indicating that WormPsyQi allows standardized quantification of synaptic features, which human annotators cannot. Importantly, our results show that the WormPsyQi model trained on one image is as good as the one trained on five, indicating that even without using a pre-trained model, WormPsyQi can save time spent on annotating synaptic images. When comparing the predictions of WormPsyQi (trained with one expert's labeling) to the labels of other experts, the IoU scores were found to be comparable to the inter-expert IoU (*Figure 2D*, columns 4 and 5).

To assess the WormPsyQi vs. human labeler effect size, we next compared the discrepancy between WormPsyQi scoring and scoring by two independent human labelers (*Figure 2—figure supplement 3*). Based on our analysis of I5p::GFP::CLA-1 and M4p::GFP::RAB-3, we found that the effect size was reporter-dependent: for I5 CLA-1, where the puncta were discrete and SNR high, inter-labeler discrepancy and WormPsyQi vs. human (labeler 1 or 2) discrepancy were low. Conversely, the discrepancy across all three scoring methods was high for M4p::GFP::RAB-3, which is a synapse-dense strain with less discrete signal; in this case, semi-automated scoring averaged out human-to-human scoring differences, but did not significantly improve the segmentation.

Lastly, we compared the quantification of additional puncta features such as puncta volume and distribution along the neurite, and overall processing time. To do so, we focused on the ASK GFP::CLA-1 reporter (*Figure 2—figure supplement 4*). We found that the difference between quantification of puncta number, volume and distribution using WormPsyQi vs. two independent human annotators is statistically insignificant, while WormPsyQi substantially reduced quantification time (*Figure 2—figure supplement 4E*). This is especially true for quantifying subtle features such as size, volume, and intensity, which may not be immediately obvious features and are much more laborious to quantify manually compared to puncta number, because pixel-wise annotation is necessary. Our analysis, therefore, suggests that the algorithm is competent in recapitulating human labeling (in some cases averaging inter-labeler variability) of not just puncta number but also other features, significantly expedites scoring, and can be used to replace manual scoring. The robustness of puncta quantification is further corroborated by comparing WormPsyQi vs. manual quantification for many strains in ensuing sections (*Figure 4—figure supplement 1*, *Figure 5*).

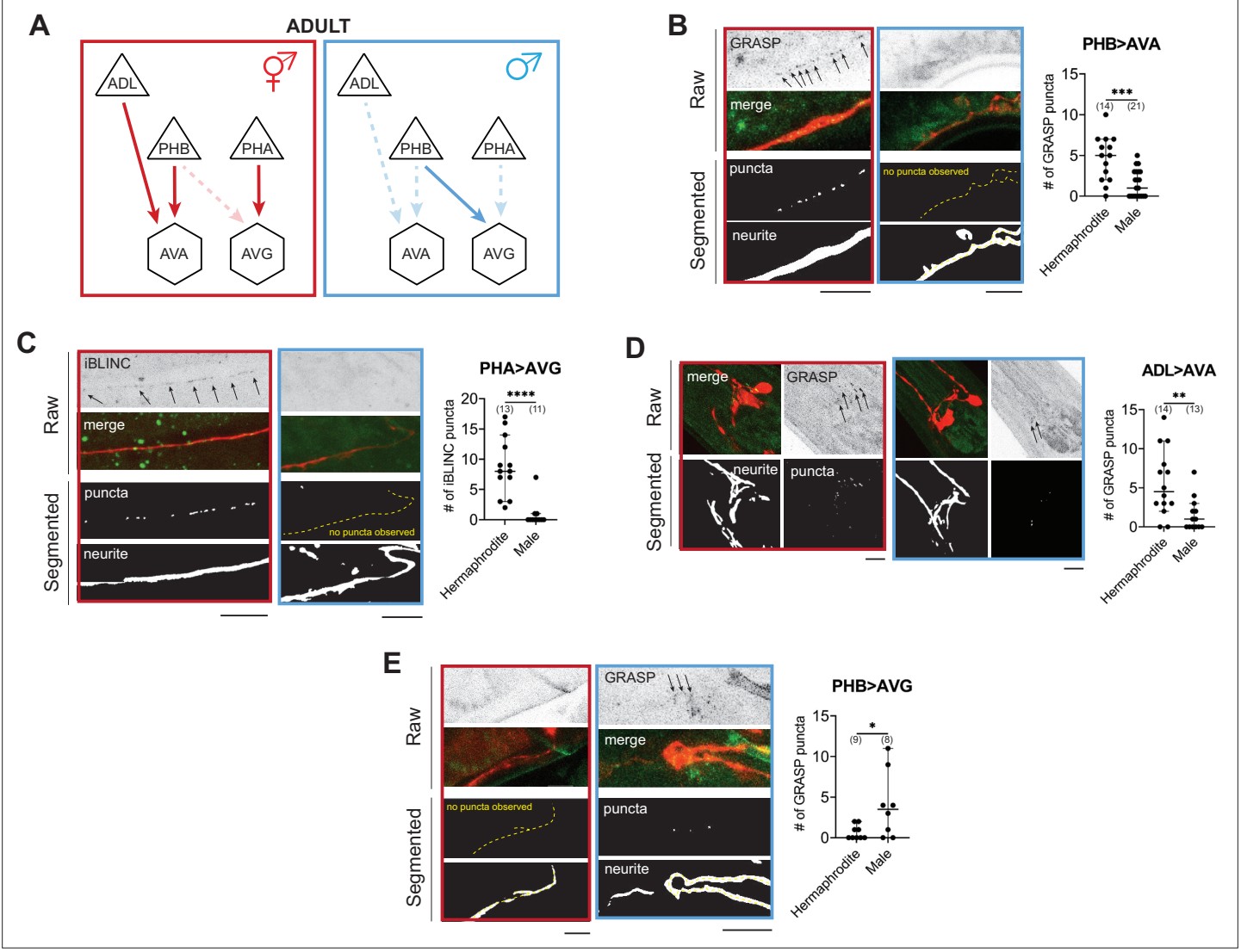

**Figure 3.** WormPsyQi validates sexually dimorphic synapses in *C. elegans*. (**A**) Subsets of hermaphrodite (red) and male (blue) connectivity diagrams, based on electron microscopy (EM) studies (***Cook et al., 2019***; ***White et al., 1986***), showing adult stage sexually dimorphic synapses analyzed for validating WormPsyQi. Synapses, depicted by arrows, were visualized using GRASP or iBLINC reporters generated either for previous studies (PHA>AVG, ADL>AVA, and PHB>AVG in ***Cook et al., 2019***; ***Bayer and Hobert, 2018***; ***Oren-Suissa et al., 2016***) or this paper (PHB>AVA). Sensory, inter-, and motor neurons are depicted as triangles, hexagons, and circles, respectively. (**B-E**) WormPsyQi validates sex-specific synapses in adult hermaphrodites and males. PHB>AVA, PHA>AVG, ADL>AVA, and PHB>AVG synapses were visualized using the transgenes *otIs839*, *otIs630*, *otEx6829*, *and otIs614*, respectively. Panels corresponding to each reporter show raw confocal images (top) and segmented neurites and synapses (bottom). Segmentation and quantification were performed using WormPsyQi. Synchronized day-1 adult animals were scored. Red – hermaphrodite, blue – male. p values were calculated using an unpaired *t*-test. ****p ≤ 0.0001, ***p ≤ 0.001, **p ≤ 0.01, and *p ≤ 0.05. In each dataset, a dot represents a single worm and lines represent median with 95% confidence interval. All raw and segmented images are maximum intensity projections of confocal Z-stacks. Scale bars = 10 μm.

## WormPsyQi validates sexually dimorphic synaptic connectivity

To demonstrate that our pipeline can quantify synaptic puncta in a robust manner, we initially set out to validate sexually dimorphic synaptic connectivity in the *C. elegans* nervous system. Whole-nervous system EM reconstruction in *C. elegans* adult males and hermaphrodites shows that many 'sex-shared' neurons – neurons present in both sexes – display sex specific, that is, sexually dimorphic synaptic connectivity patterns (***Cook et al., 2019***). In previous studies, a small subset of these dimorphic synapses was visualized with GRASP- or iBLINC-based transgenic reporters (***Pechuk et al., 2022***; ***Salzberg et al., 2020***; ***Cook et al., 2019***; ***Weinberg et al., 2018***; ***Bayer and Hobert, 2018***; ***Oren-Suissa***

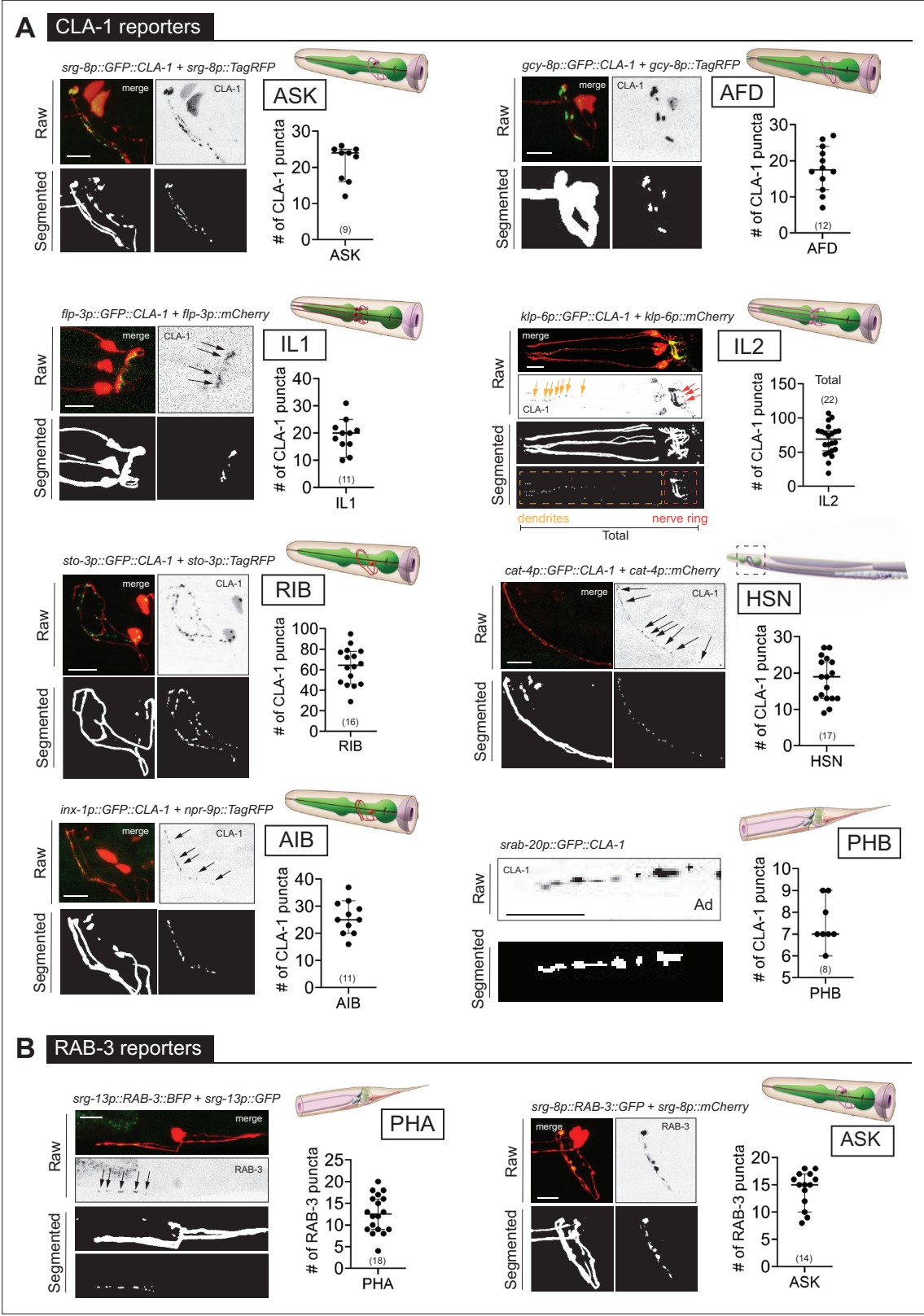

**Figure 4.** WormPsyQi is generalizable across diverse synaptic reporters targeting the synapses in the central nervous system. (**A**) Representative images of cell-specific synaptic reporters showing presynaptic specializations, as visualized with GFP-tagged CLA-1, for neuron classes ASK (*otIs789*), AFD (*otEx7786*), IL1 (*otEx7363*), IL2 (*otIs815*), RIB (*otIs810*), HSN (*otIs788*), AIB (*otEx8023; otIs886*), and PHB (*otIs883*). (**B**) Representative images of cell-specific synaptic reporters showing presynaptic specializations, as visualized with fluorescently tagged RAB-3, for neuron classes PHA (*otIs702*) and ASK

*Figure 4 continued on next page*

*Figure 4 continued*

(*otEx7231*). In each panel, top (raw) and bottom (segmented) images represent both neuronal processes, based on a cytoplasmic marker, and synaptic puncta. There was no cytoplasmic marker in the PHB reporter used, so only puncta are shown. All images are maximum intensity projections of confocal Z-stacks. Quantification of the number of puncta, performed using WormPsyQi, is shown in corresponding graphs. In each graph, a dot represents a single worm and lines represent median with 95% confidence interval. L4 animals were scored unless otherwise noted. Scale bars = 10 μm.

The online version of this article includes the following figure supplement(s) for figure 4:

**Figure supplement 1.** WormPsyQi performs optimally on images with a cytoplasmic marker and discrete puncta.

et al., 2016). By scaling the analyses, presently extremely laborious with EM, these reporters have been indispensable for not only substantiating EM data, but also for understanding the molecular determinants underlying dimorphic circuits and the pathways mediating plasticity (*Goodwin and Hobert, 2021* and references therein). In addition, the ability to visualize a specific synapse across many animals continues to reveal new insights such as the extent of variability in synaptic number and spatial distribution across animals, which was previously unappreciated in the absence of tools and sufficient EM samples to study it.

To validate these findings using WormPsyQi, we honed in on four synaptic connections between sex-shared neurons: PHB>AVA, PHA>AVG, PHB>AVG, and ADL>AVA. In all cases, previous studies have shown sex-specific differences in these connections after sexual maturation (*Figure 3A*; *Cook et al., 2019*; *Oren-Suissa et al., 2016*). We show here that WormPsyQi reliably validates previous observations: synapses between ADL>AVA, PHB>AVA, and PHA>AVG are sexually dimorphic, such that synapse number in adult males is less compared to hermaphrodites, and PHB>AVG synapses are less in adult hermaphrodites compared to males (*Figure 3B–E*). An added advantage compared to previous manual puncta quantification was that WormPsyQi expedited the quantification by largely automating the process and, importantly, relied on a single pre-trained segmentation classifier to robustly quantify synaptic number across all four reporters. Together, the performance on these test cases encouraged us to further explore the efficacy of our pipeline in a more systematic manner.

## An expanded toolkit to visualize synapses in *C. elegans*

Having established that WormPsyQi works on a handful of synaptic reporters, we sought to further explore its efficacy in facilitating unbiased synapse quantification. The aforementioned sexually dimorphic reporters (*Figure 3*) are all GRASP- or iBLINC-based and target only five neuron classes – most making synapses in the *C. elegans* tail – of the total 116 sex-shared classes in hermaphrodites (*White et al., 1986*). We asked if our pipeline could quantify various types of punctate signals across other reporter types and in other regions of the nervous system, particularly the nerve ring, which comprises the bulk of the hermaphroditic *C. elegans* connectome (*Brittin et al., 2021*; *Cook et al., 2019*; *White et al., 1986*). Despite this preponderance, the nerve ring has been remarkably under-studied largely due to a lack of tools available to visualize and reliably quantify synapses in the most complex and synapse-dense region of the *C. elegans* nervous system. This is generally true for synapses in the somatic or central nervous system (CNS), compared to neuromuscular junctions, which are easier to visualize and have more available reporters.

To broaden the existing resource of synaptic reporter strains, we generated novel synaptic marker strains. These include cell-specific CLA-1 reporters for eight neuron classes (*Figure 4*) and five new GRASP reporters (*Figure 5*) which label specific synapses between multiple neuron classes. Considering both new and previously existing reporters, we describe synaptic reporters for almost 30 neuron classes in both the central and pharyngeal (=enteric nervous systems) of *C. elegans* (*Figure 1*; *Supplementary file 1*). Many of the new reporters described here target neurons that make synapses in the nerve ring, including cell-specific reporters for IL1, ASK, ADL, RIB, AIA, BDU, HSN (first published in *Leyva-Díaz and Hobert, 2022*), and IL2 (first published in *Cros and Hobert, 2022*; *Figure 4*). These efforts relied on the presence of cell-specific promoters driving GFP expression. The recent availability of single-cell RNA-sequencing (scRNAseq) data driven by the CeNGEN project (*Taylor et al., 2021*) opens up opportunities to gain even broader coverage.

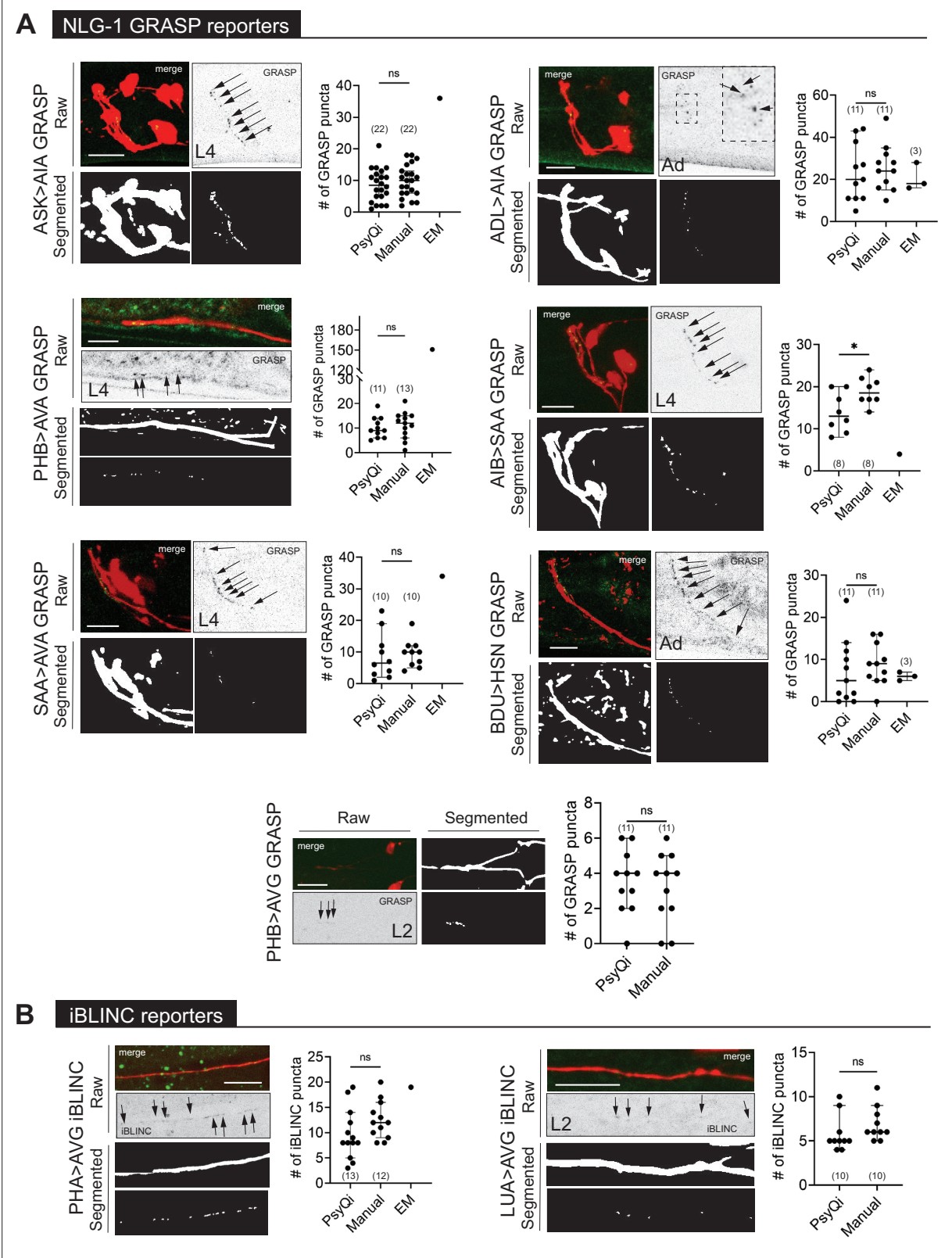

**Figure 5.** WormPsyQi can quantify synapses in variable GRASP and iBLINC reporters. (**A**) Representative images of NLG-1 GRASP-based synaptic reporters labeling the following synapses: ASK>AIA (*otIs653*), ADL >AIA (*otEx7457*), PHB >AVA (*otIs839*), AIB-SAA (*otEx7809*), SAA-AVA (*otEx7811*), BDU>HSN (*otEx7759*), and PHB>AVG (*otIs614*). (**B**) Representative images of iBLINC-based synaptic reporters labeling PHA>AVG (*otIs630*) and LUA>AVG (*otEx6344*) synapses. WormPsyQi vs. manual quantification is shown in corresponding plots; electron microscopy (EM) data are shown where

*Figure 5 continued on next page*

*Figure 5 continued*

available (*Witvliet et al., 2021*; *Cook et al., 2019*; *White et al., 1986*), but was excluded from statistical analyses due to small sample sizes (*n* = 1–3). In each dataset, a dot represents a single worm and lines represent median with 95% confidence interval. All raw and segmented images are maximum intensity projections of confocal Z-stacks. p values were calculated using an unpaired *t*-test. *p ≤ 0.05 and p > 0.05not significant (ns). Scale bars = 10 μm.

## WormPsyQi is generalizable across a wide array of synaptic reporters in the CNS

Previous synapse quantification approaches in *C. elegans* focused on neurons with relatively simple trajectories along the anterior–posterior axis in one plane on the dorso-ventral axis (e.g. *en passant* synapses along the ventral nerve cord). These include semi-automated (*San-Miguel et al., 2016*; *Crane et al., 2012*) and manual approaches such as counting synaptic puncta by hand or drawing plot profiles (line scans) of fluorescent signals along a neurite using open-source image processing software such as Fiji/ImageJ, often combined with synapse enrichment quantification as assessed by fluorescence (*Kurshan et al., 2018*; *Xuan et al., 2017*; *Jang et al., 2016*; *Dittman and Kaplan, 2006*). These methods are well suited for quantifying synapses along linear neurites, but pose problems in neurons with morphologically complex anatomies in 3D space. We thus wanted our pipeline to be able to easily quantify synapses in other types of neurons in *C. elegans*.

In addition, a key observation based on many synaptic reporters imaged by different researchers using diverse microscopes, and across different days, life stages, and sexes, is that images collected from these reporters are perceptibly heterogeneous depending on the type of fluorophore-tagged synaptic protein, choice of promoter driving the fluorophore, and the labeling method used. For instance, cell-specific RAB-3 reporters have a more diffuse synaptic signal compared to the punctate signal in CLA-1 reporters for the same neuron, as shown for the neuron pair ASK (*Figure 4—figure supplement 1C*) and observed previously in other neurons (*Cook et al., 2020*; *Lipton et al., 2018*; *Xuan et al., 2017*). This is intuitive based on the distinct localization and function of RAB-3, a presynaptic vesicle protein (*Nonet et al., 1997*), and CLA-1/Piccolo, an active zone scaffolding protein that recruits synapse vesicles to release sites (*Xuan et al., 2017*). We note that this is not a consequence of tagging RAB-3 on a specific terminal, since both N- and C-terminal RAB-3 fusions manifest more diffuse signals compared to CLA-1. While this may not be a problem for neurons or regions with low synapse density (e.g. ASK RAB-3 and CLA-1 puncta number are similar as shown in *Figure 4—figure supplement 1C*), it poses challenges for synapse-dense cases such as AIB and SAA, where WormPsyQi is unable to reliably segment puncta using any of the four pre-trained models. Lastly, within a reporter type, fluorescent signals appear similar, as demonstrated by the similarity of punctate fluorescent signal between various CLA-1 reporters (*Figure 4A*). This is useful because it allows the use of a single pre-trained model for multiple use cases.

In contrast to the relative signal homogeneity in presynaptic fluorophore-tagged reporters of the same kind, signals across reporters that make use of transsynaptic technologies can vary. Fluorescent signals from GRASP and iBLINC reporters appear different from other reporters and is wide-ranging across neurons or synapses (*Figure 5*). As previously reported, this is likely due to transsynaptic labeling methods being more sensitive to the choice of promoters used to drive split-fluorophore fragments, amount of injected reporter DNA, extrachromosomal vs. integrated nature of transgenes, and imaging conditions (*Oren-Suissa et al., 2016*). Consequently, quantifying the signal of such reporter lines has been challenging and requires expert knowledge, which is not always transferable across researchers and labs introducing human bias and error.

To address these concerns, we used WormPsyQi to quantify puncta in representative datasets for many different presynaptic (*Figure 4*) and transsynaptic (*Figure 5*) reporters for neurons in the CNS. We first trained reporter type-specific synapse classifiers (see Methods) to aid quantification so that an average user does not need to perform the tedious step of labeling and training datasets for common types of reporters available in the community. Next, we systematically ran the image datasets through our pipeline, making use of neurite masking (where available) prior to synapse segmentation and quantification (*Figure 2—figure supplement 1*; *Figures 4 and 5*). Analyzing many reporters further validated that WormPsyQi was able to detect synaptic puncta with minimal human contribution and across a wide range of reporter types (*Figures 4 and 5*).

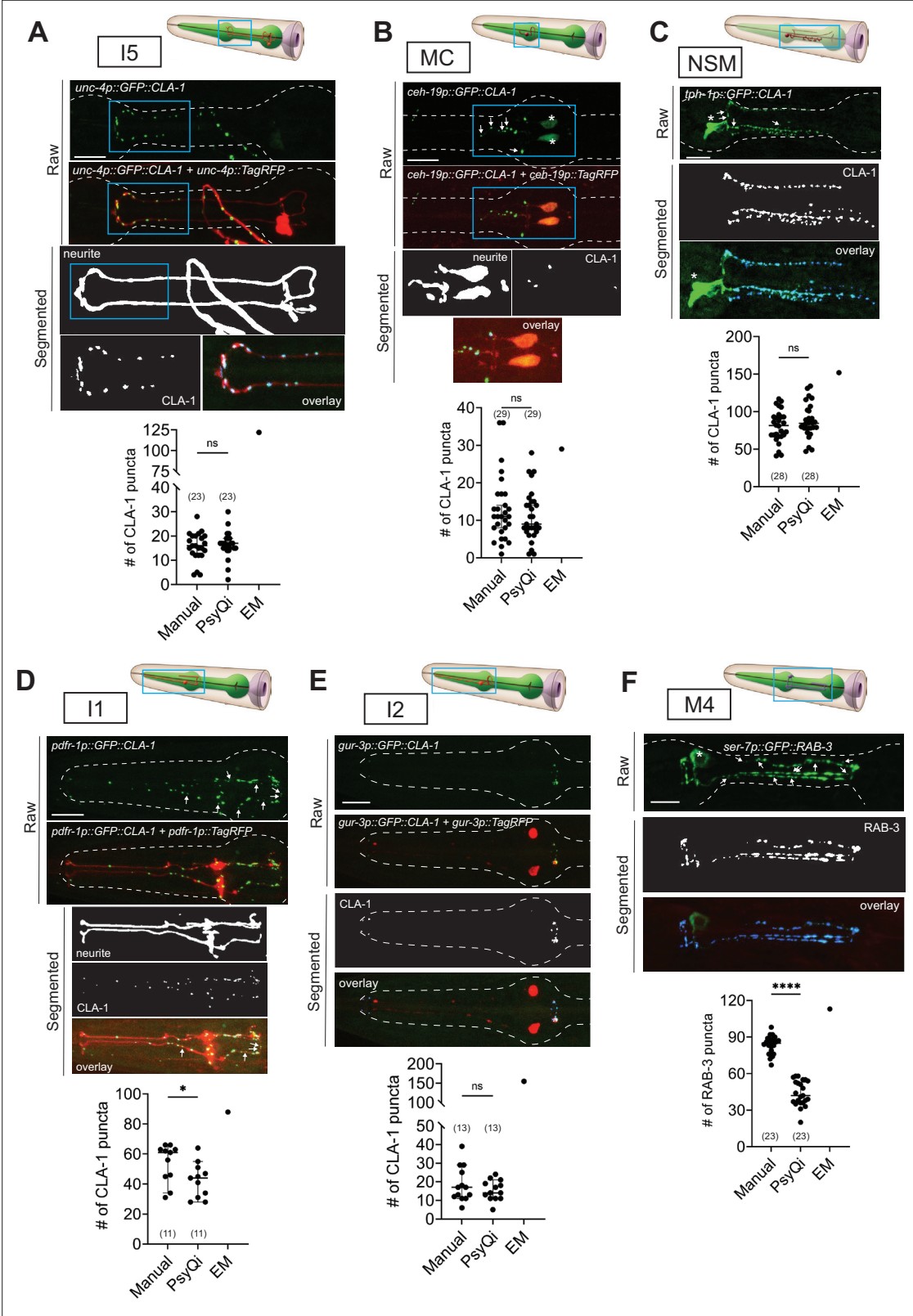

**Figure 6.** Systematic analysis of pharyngeal synapses using WormPsyQi. CLA-1- or RAB-3-based presynaptic specializations of pharyngeal neurons: (**A**) I5, (**B**) MC, (**C**) NSM, (**D**) I1, (**E**) I2, and (**F**) M4 analyzed using transgenes *otEx7503*, *otEx7505*, *otEx7499*, *otEx7501*, and *otIs597*, respectively. For each reporter, a corresponding graph shows manual vs. WormPsyQi quantification, along with electron microscopy (EM) synapse counts as described in *Cook et al., 2020*. Day-1 adults were scored, with some datasets previously published (*Cook et al., 2020*) and re-analyzed here. Masking was used

*Figure 6 continued on next page*

*Figure 6 continued*

prior to synapse prediction for all neurons except I2, where the cytoplasmic reporter was too dim for creating a continuous mask. Puncta that were not segmented by WormPsyQi are marked with white arrows in the raw GFP image to help inform users what kind of images/reporters are best quantified using WormPsyQi. Cell bodies are marked with an asterisk. p values were calculated using an unpaired *t*-test. ****p ≤ 0.0001, *p ≤ 0.05, and p > 0.05 not significant (ns). EM data were excluded from statistical analyses based on limited sample size (*n* = 1). In each dataset, a dot represents a single worm and lines represent median with 95% confidence interval. All raw and segmented images are maximum intensity projections of confocal Z-stacks. Scale bars = 10 µm.

The online version of this article includes the following figure supplement(s) for figure 6:

**Figure supplement 1.** WormPsyQi analysis of all presynaptic specializations in the pharynx.

To confirm that WormPsyQi quantified fluorescent puncta accurately, we compared its results with manual quantification for a subset of reporters analyzed and found that in most datasets, the difference between WormPsyQi and human annotation was statistically insignificant (***Figure 4—figure supplement 1A***, ***Figure 5***). In cases where our pipeline could not recapitulate manual scoring, the masking step was either skipped prior to segmentation (PHB CLA-1) or synapse density was too high (IL1 CLA-1). In the case of IL1-expressed CLA-1, training a reporter-specific classifier improved the results (***Figure 4—figure supplement 1***). Generally, having a cytoplasmic reporter in the background of the synaptic reporter enhanced performance; it was also a crucial step in preventing synapse mis-annotation in regions with high background fluorescence intensity such as the tail, where adjacent intestinal autofluorescence typically masks bona fide synaptic signal. This is demonstrated in the case

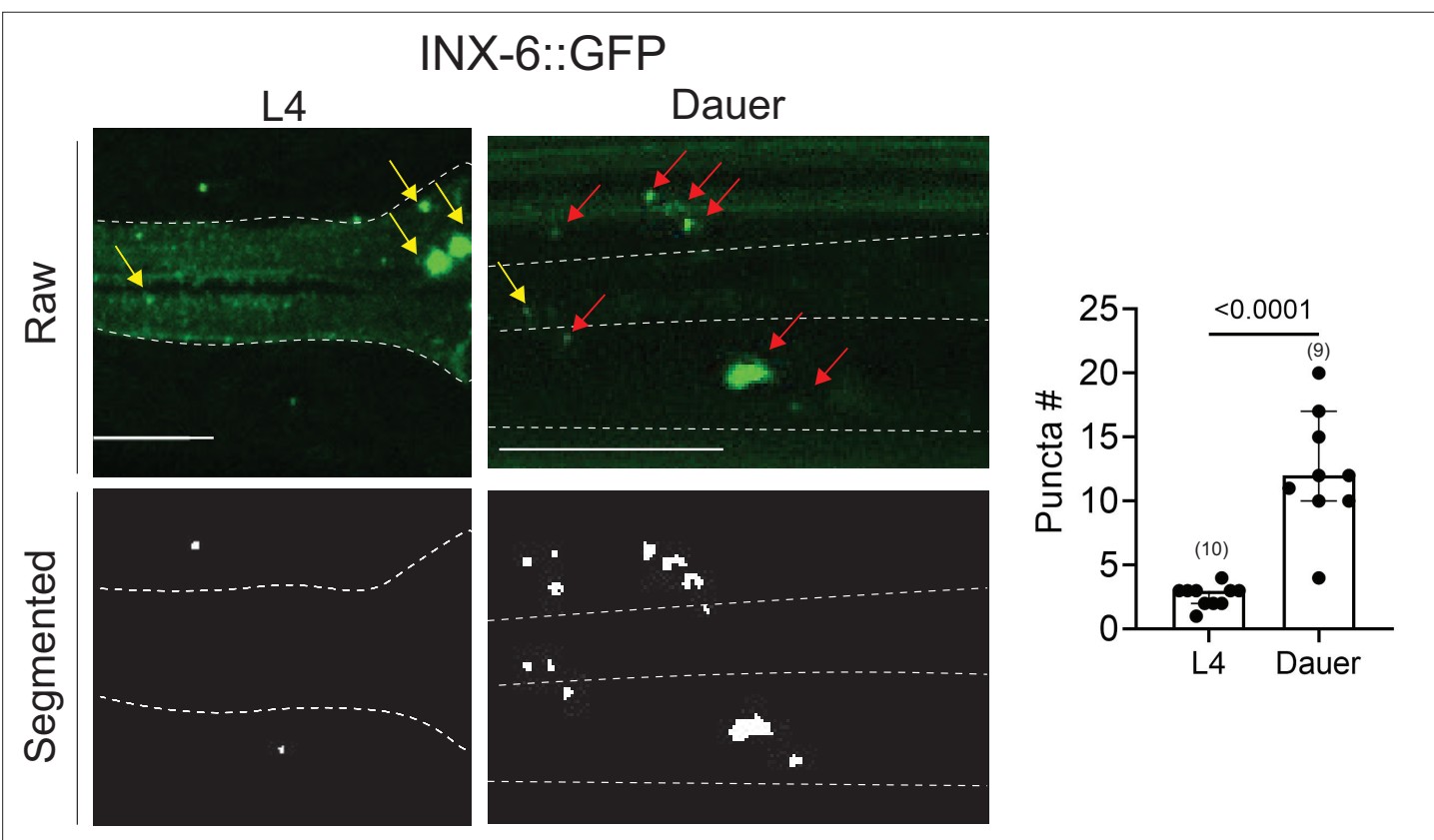

**Figure 7.** An analysis of electrical synapses using WormPsyQi. INX-6::GFP (*ot805*) puncta are observed in the pharynx and the nerve ring. Nerve ring gap junctions are compared between L4 and dauer animals to validate previous observations (***Bhattacharya et al., 2019***). Given that outside of the pharynx, *inx-6* is exclusively expressed in AIB interneurons in dauers, the puncta can be assigned to AIB axons as shown previously (***Bhattacharya et al., 2019***); in the absence of a cytoplasmic marker, WormPsyQi scores a limited number of puncta in L4 animals, but the dauer count significantly exceeds L4 count, as expected. Red arrows denote AIB-localizing electrical synapses and yellow arrows denote electrical synapses in the pharyngeal muscles. All raw and processed images are maximum intensity projections of confocal Z-stacks. Statistical analysis was performed using an unpaired *t*-test. Nerve ring synapses are denoted by a yellow box; synapses outside the box are pharyngeal. In each dataset, a dot represents a single worm and the height of the bar represents median with 95% confidence interval. Images are maximum intensity projections of confocal Z-stacks. Scale bars = 10 µm.

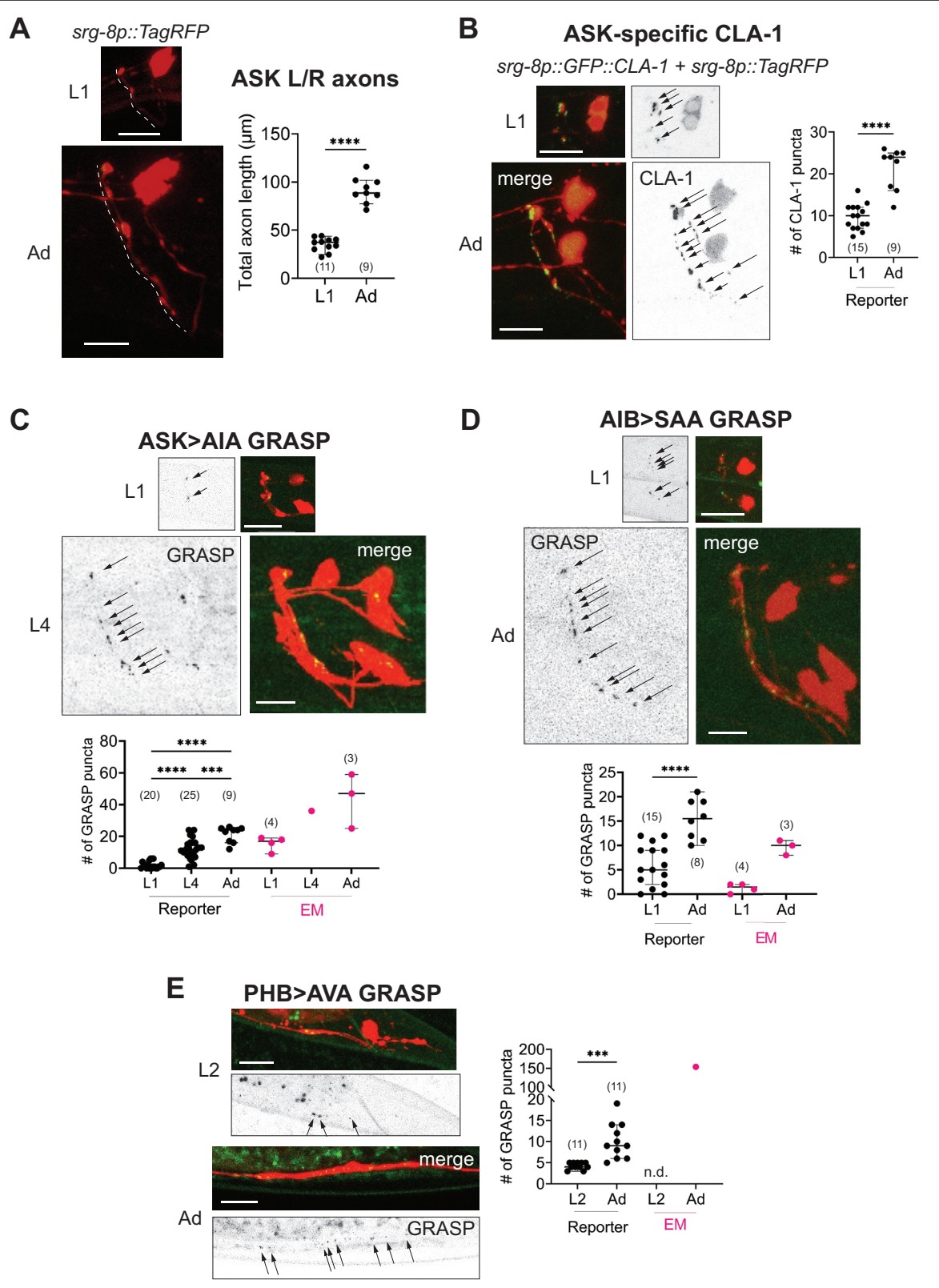

**Figure 8.** Developmental synapse addition in the central nervous system (CNS) roughly scales with neurite length in strains analyzed. (**A**) Representative quantification of the total nerve ring length of late-L1 and day-1 adult animals, as analyzed with an ASK-specific cytoplasmic transgene (*srg-8p::tagRFP* in reporter *otIs789*). Total ASK axon length in the nerve ring increases ~twofold between late-L1 and adulthood. Dashed lines mark axons in laterally positioned worms. (**B**) Representative images of an ASK-specific CLA-1 reporter (analyzed using the transgene *otIs789*) at late-L1 and day-1 adult

*Figure 8 continued on next page*

*Figure 8 continued*

stages. The graph shows the number of puncta representing ASK-specific presynaptic specializations quantified using WormPsyQi. The number of puncta increases ~twofold, which is in proportion with the increase in axon length across development. Representative images and WormPsyQi-based quantification of synapse addition using synapse-specific GRASP reporters visualizing (**C**) ASK>AIA (*otIs653*), (**D**) AIB>SAA (*otEx7809*), and (**E**) PHB>AVA (*otIs839*) synapses at early larval (L1 or L2) and L4 or day-1 adult (Ad) stages. p values were calculated using an unpaired *t*-test (**A, B, D, E**) or one-way analysis of variance (ANOVA) with Bonferroni correction for multiple comparisons (**C**). ****p ≤ 0.0001 and ***p ≤ 0.001. Electron microscopy (EM) data (taken from *Witvliet et al., 2021*; *Cook et al., 2019*; *White et al., 1986*) were excluded from statistical analysis based on limited sample sizes (n = 1–4), which were out of proportion with reporter-based data. In each dataset, a dot represents a single worm and lines represent median with 95% confidence interval. All raw and segmented images are maximum intensity projections of confocal Z-stacks. Scale bars = 10 μm.

of PHB>AVA GRASP puncta quantification (*Figure 4—figure supplement 1B*), where WormPsyQi erroneously segments gut autofluorescence puncta as synapses in the absence of a cytoplasmic reporter.

We also found that CLA-1 reporters were overall a superior marker compared to RAB-3 reporters for the same neuron, as shown for ASK (*Figure 4—figure supplement 1A, C*). In both cases, Worm-PsyQi could segment puncta, but the signal was much more discrete, and synapses were resolved better in the ASK CLA-1 reporter. In the case of ASK, overall puncta number in L4 animals was similar in both RAB-3 and CLA-1 reporters but the mean puncta area was much greater in the case of RAB-3, corroborating the diffuse signal, which poses quantification challenges for neurons with greater synapse densities, for example AIB (*Figure 4—figure supplement 1D*). We also note that the difference in diffuse vs. discrete signal could not be attributed to the transgenic reporter type (integrant vs. extrachromosomal) since both types of CLA-1 reporters had discrete, smaller puncta compared to RAB-3; owing to variable expressivity in the CLA-1 extrachromosomal reporter, however, fewer puncta could be quantified (*Figure 4—figure supplement 1C*). Lastly, in some reporters (*Figure 4—figure supplement 1D*), such as those for RME-specific SNB-1, SAA-specific RAB-3, and AIB-specific RAB-3, WormPsyQi could not effectively segment synapses because the signal was too diffuse and synapse density, in the case of AIB, too high. Again, relying on alternative reporters where possible, such as the CLA-1-based reporter for AIB (*Figure 4A*), which is much better resolved and has more distinguishable puncta, is recommended.

In some cases (e.g. AIB>SAA in *Figure 5*), there were discrepancies between WormPsyQi and manual quantification because the signal intensity of puncta was too low to be quantified comprehensively using our pipeline. In cases where these issues cannot be resolved during data acquisition, further improvement may be achieved by using alternative microscopy methods, reporter types, or brighter fluorophores. We also note that in several cases, GRASP quantification differed from EM scoring (*Figure 5*). In the absence of a comparable EM sample size in any paradigm tested, it is impossible to assess whether the difference is statistically significant. We note that differences between light-microscopy-based imaging and EM are anticipated particularly in synapse-dense regions where our fluorescent reporter-based imaging cannot resolve individual synapses as well as EM analysis does.

## WormPsyQi performs robustly in morphologically diverse pharyngeal neurons

In addition to the somatic or CNS described thus far, *C. elegans* contains a pharyngeal or enteric nervous system (ENS) which resides entirely in the pharynx (*Avery and You, 2012*; *Mango, 2007*; *Albertson and Thomson, 1976*). Unlike the CNS, the dynamics of synaptic connectivity in the ENS are even less explored. Previous EM reconstructions of the pharynx (*Cook et al., 2020*; *Albertson and Thomson, 1976*) serve as excellent starting points but, again, we are confronted with limitations such as a very small sample size and a lack of means to study how synapses in the pharynx are established and altered in larval development, as well as under various perturbations.

The *C. elegans* pharynx constitutes 14 neuron classes, and like the ENS in other organisms (*Brehmer, 2021*; *Furness, 2007*; *Costa et al., 2000*), there is substantial morphological heterogeneity among these neuron classes (*Albertson and Thomson, 1976*). In addition, some neurons show variability in branching and synapses, both within a bilateral pair and across individuals, such as in the case of M3 L/R pair (*Albertson and Thomson, 1976*). The diversity and variability in form pose an additional challenge for image processing, particularly in masking neurites using conventional software. We thus

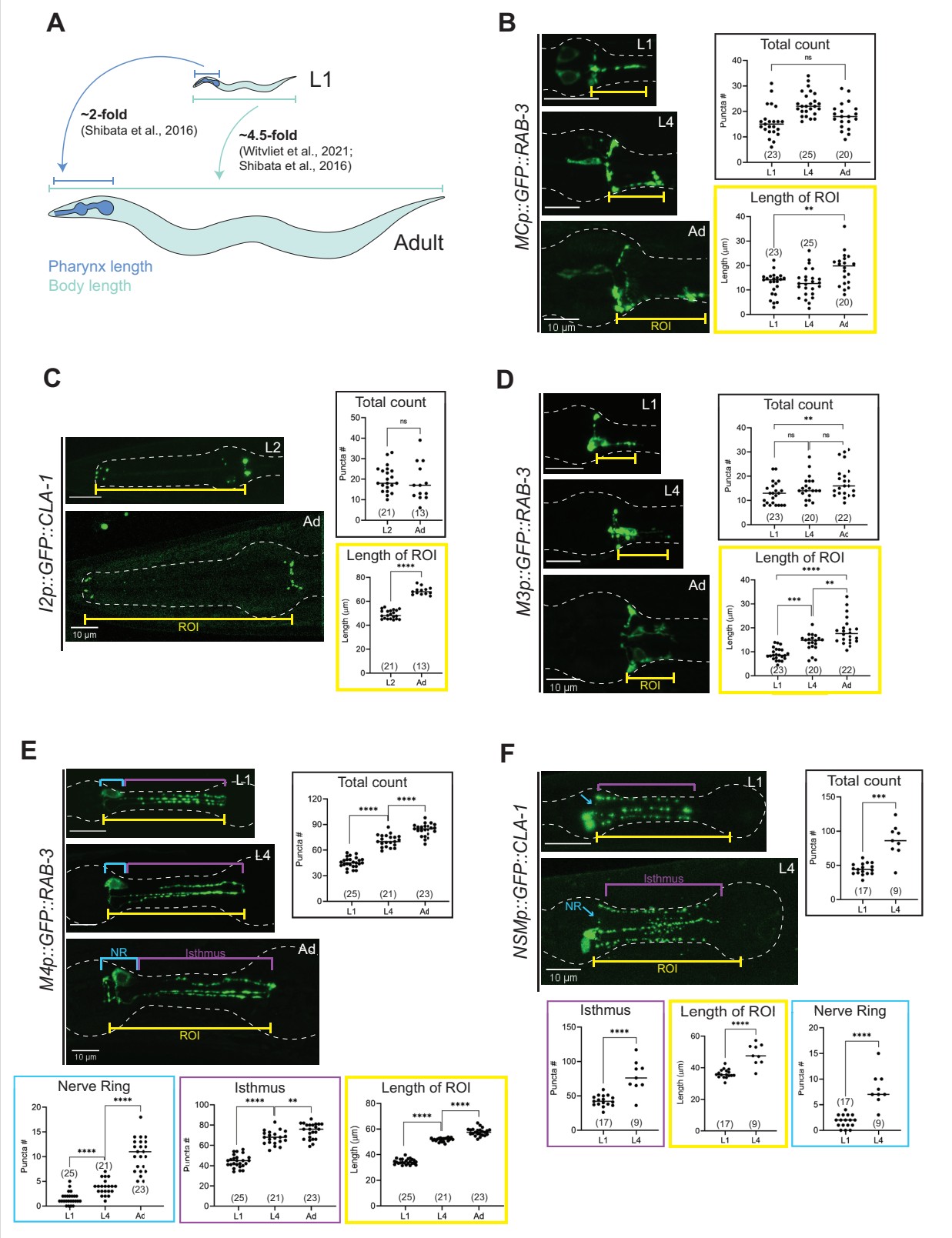

**Figure 9.** Synapse addition in the enteric nervous system does not scale with neuron growth over developmental. (**A**) Overview of pharynx and total body length elongation between L1 and adult stages based on previous findings (*Witvliet et al., 2021*; *Shibata et al., 2016*). Representative images and quantification of number of presynaptic specializations at early larval and adult stages visualized using (**B**) GFP::RAB-3 in MC (*otEx7505*), (**C**) GFP::CLA-1 in I2 (*otEx7497*), (**D**) GFP::RAB-3 in M3 (*otIs602*), (**E**) GFP::RAB-3 in M4 (*otIs597*), and (**F**) GFP::CLA-1 in NSM (*otEx7499*) neurons. Manual

*Figure 9 continued on next page*

*Figure 9 continued*

counts of total CLA-1 or RAB-3 puncta are plotted for each reporter (black boxes). For neurons MC, M3, M4, and NSM, region of interest (ROI, yellow box) length was measured from the pharyngeal nerve ring to the most posterior CLA-1 or RAB-3 signal observed. For I2 neurons, ROI length was measured from the pharyngeal nerve ring to the tip of the anterior end of the pharynx based on brightfield. For NSM neurons, ROI length was measured from the pharyngeal nerve ring to the grinder in the terminal bulb, based on brightfield. For neurons M4 and NSM, additional quantification was performed for the pharyngeal nerve ring region (NR, blue box) and the isthmus (purple). p values were calculated using Mann–Whitney test. ****p ≤ 0.0001, ***p ≤ 0.001, **p ≤ 0.01, and p > 0.05 not significant (ns). In each dataset, a dot represents a single worm and lines represent median. All images are maximum intensity projections of confocal Z-stacks. Scale bars = 10 µm.

The online version of this article includes the following figure supplement(s) for figure 9:

**Figure supplement 1.** Manual and WormPsyQi quantification show the same developmental trends in synapses in the pharynx.

**Figure supplement 2.** WormPsyQi quantification reveals small but significant differences between adjacent developmental stages.

asked if WormPsyQi could create neurite masks of diverse pharyngeal reporters, and subsequently score colocalized synaptic puncta. Cell-specific CLA-1::GFP synaptic reporters of several pharyngeal neuron classes were recently developed, namely, for NSM, I1, I2, I5, and MC (*Cook et al., 2020*). In addition, we describe here two RAB-3::GFP reporters which label presynaptic RAB-3 in neurons M3 and M4 to altogether cover seven neuron classes in the pharynx (*Figures 6 and 9*; *Supplementary file 1*). We also analyzed a pan-pharyngeal reporter in which presynaptic CLA-1 in all 14 pharyngeal neuron classes is visualized with GFP (*Figure 6—figure supplement 1*; *Vidal et al., 2022*).

Our analyses show that WormPsyQi can robustly segment pharyngeal neurites and generally performs well in quantifying synapses, as with the CNS. This is shown by the comparison between manual and WormPsyQi counts for each reporter (*Figure 6A–F*). In reporters for neurons I5, MC, NSM, and I2, we found no significant difference between puncta count measured by the pipeline vs. a human annotator (*Figure 6A–D*). In two reporters, there was a discrepancy between WormPsyQi and human quantification. In the case of the M4 RAB-3 reporter, the RAB-3 signal was too diffuse to resolve all puncta (*Figure 6F*, *Figure 6—figure supplement 1A*); in reporters for neurons with high synapse densities, such as I1-specific CLA-1 reporter (*Figure 6D*) and pan-pharyngeal CLA-1 reporter (*Figure 6—figure supplement 1B*), some puncta were not detected by the pipeline. In the case of the pan-pharyngeal reporter, WormPsyQi detected most puncta consistently (reflected in varying standard deviation: SD = 4.2 for posterior bulb, SD = 29.2 for total count) in the synapse-sparse posterior bulb compared to overall puncta for all pharyngeal neurons (*Figure 6—figure supplement 1B*). We expect that the presence of a sparse cytoplasmic marker in the background – which would allow using sparse neurite masking before synapse segmentation in WormPsyQi – or alternative labeling strategies would improve results. Finally, we note that there is a marked discrepancy between puncta count in our reporters compared to EM data (*Figure 6*), but due to limited EM sample size, such differences are difficult to interpret.

## WormPsyQi can quantify and scale the study of electrical synapses

Although connectomics has historically been biased toward chemical synapses, expression studies on the localization of key electrical synapse components – innexins in invertebrates and connexins/pannexins in vertebrates – suggest the presence of complex electrical connectivity patterns (*Ammer et al., 2022*; *Bhattacharya et al., 2019*; *Feigenspan et al., 2004*; *Lee et al., 2003*). This is strengthened by evidence of gap junctions in ultrastructural EM-based analyses in both invertebrates (*Cook et al., 2019*; *Jarrell et al., 2012*; *White et al., 1986*) and vertebrates (*Smedowski et al., 2020*; *Anderson et al., 2011*), as well as expression studies showing cell type- and tissue-specific spatial patterns of innexin and connexin genes (*Bhattacharya et al., 2019*). In addition, studies involving perturbations of innexins and connexins have shown diverse contributions of electrical synapses to neuronal function (*Hendi et al., 2022*; *Bhattacharya et al., 2019*; *Hall, 2017*; *Song et al., 2016*; *Meng et al., 2016*; *Kawano et al., 2011*; *Bloomfield and Völgyi, 2009*; *Yeh et al., 2009*; *Chuang et al., 2007*).

Scoring electrical synapses using EM has proven more challenging (*Emmons et al., 2021*). However, non-EM-based approaches to study gap junctions are readily available since electrical synapses can be directly visualized by endogenously tagging junction-forming innexins, which assemble into hemichannels in pre- and post-synaptic regions before forming gap junctions via complementary matching between opposing hemichannels (*Figure 1*; *Goodenough and Paul, 2009*).

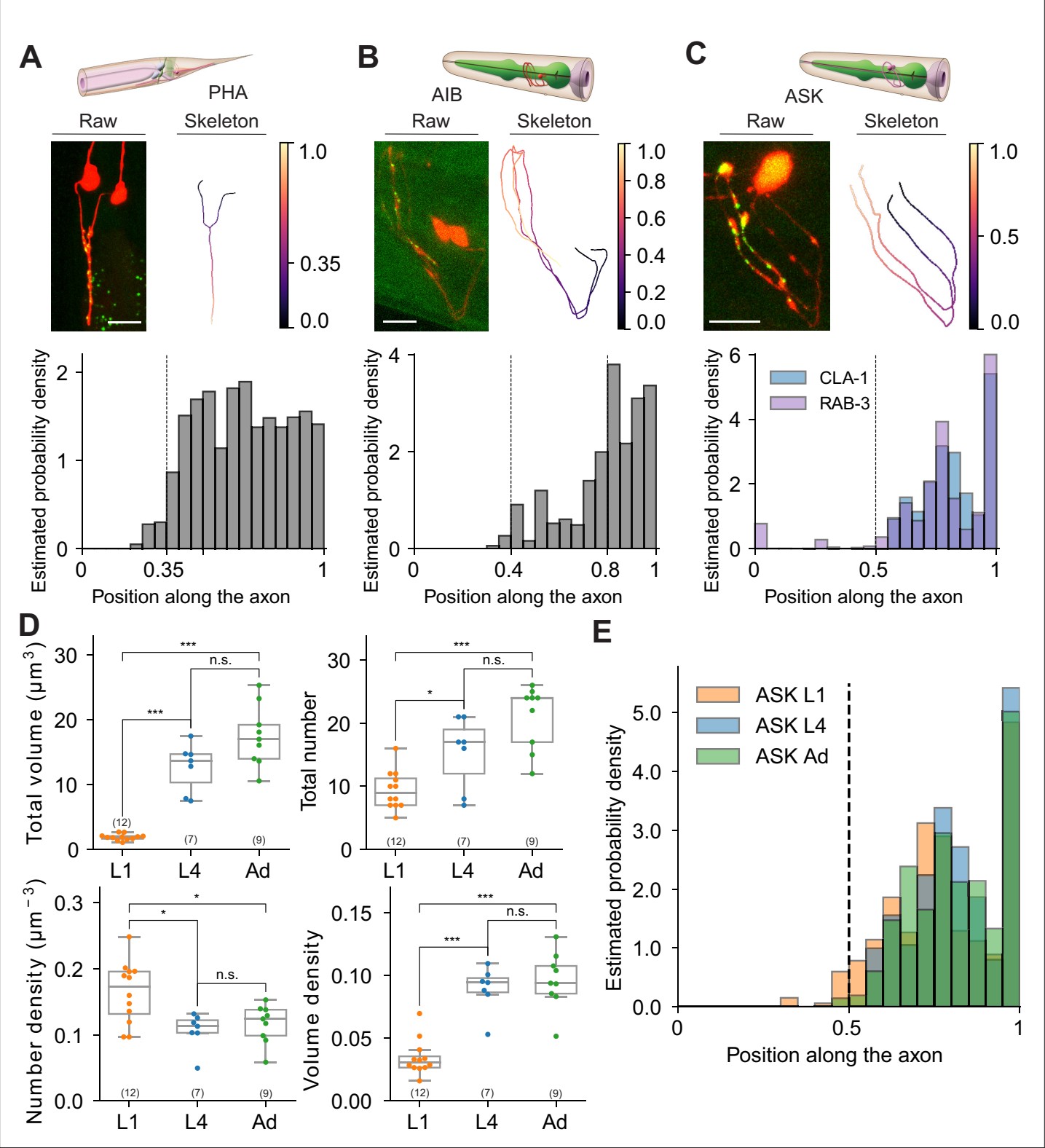

**Figure 10.** Extracting subcellular and spatial synaptic features using WormPsyQi. (**A–C**) Synaptic puncta distribution profiles of RAB-3 in PHA, CLA-1 in AIB, and CLA-1 and RAB-3 in ASK neurons. Top: schematic of neurons (source: WormAtlas). Middle left: maximum intensity projection of raw image with contrast enhancement. Middle right: skeleton diagram of neurons. Position 0 indicates soma and 1 indicates the furthest axon terminal. Bottom: probability density histogram of puncta distribution along the neurite. Puncta distribution profiles of worms at the same stage were combined to give a histogram (see Methods). (**C**) RAB-3 and CLA-1 puncta profiles overlaid. (**D**) Representative features describing developmental changes of ASK's

*Figure 10 continued on next page*

*Figure 10 continued*

presynaptic specializations. Top: puncta volume and puncta number per worm, from left to right. Bottom: puncta number density and puncta volume density per worm (calculated along the neurite domain covered where synapses are localized). p values were calculated using one-way analysis of variance (ANOVA) with Bonferroni correction for multiple comparisons. ***p ≤ 0.001, *p ≤ 0.05, and p > 0.05 not significant (ns). In each dataset, a dot represents a single worm. (**E**) Synaptic puncta density distribution in ASK neurons at different stages. For each worm, the synaptic volume distribution was normalized by axon length.

The online version of this article includes the following figure supplement(s) for figure 10:

**Figure supplement 1.** Distribution of CLA-1 or RAB-3 puncta in individual worms.

**Figure supplement 2.** Extended features and representative two-dimensional scatter plots of ASK CLA-1 reporter across different developmental stages.

To analyze datasets for electrical synapses using WormPsyQi, we focused on a sparsely expressed innexin reporter, INX-6::GFP (*Bhattacharya et al., 2019*; *Figure 7*). WormPsyQi was able to segment and quantify puncta using the default classifier (based on CLA-1 trained images). INX-6::GFP puncta were observed in the pharynx and the nerve ring region, with more puncta scored in dauers compared to well-fed L4 animals (*Figure 7*). A subset of these puncta represents *de novo* gap junctions between AIB and BAG neurons which mediate dauer-specific chemosensory behavior (*Bhattacharya et al., 2019*).

## Quantitative analysis of synapses across development in the CNS

Synapse plasticity takes multiple forms – such as synapse addition at pre-existing synaptic sites, pruning, and establishment of *de novo* synapses – and is essential for shaping neuronal circuits and behavior. Most studies describing synapse addition use EM-based approaches and rely on a small sample size. An obvious pitfall of this is that resulting low-throughput datasets preclude distinguishing synapse plasticity across life stages from baseline inter-individual variability. While expanding EM analysis to study larger sample sizes would be an ideal solution, simply relying on transgenic reporters and the high-throughput analyses they enable could facilitate studying synaptic plasticity in light of variability.

In this paper, we used our toolkits to explore how synaptic puncta change in a small subset of neurons by comparing the number of puncta between early larval and adult stages (*Figure 8*). From our analysis, we found that generally neurons in the CNS progressively add synapses at pre-existing sites throughout larval development up until adulthood (*Figure 8B–E*). For synapses studied in detail (ASK>AIA, PHB>AVA, and AIB>SAA visualized with GRASP, and ASK CLA-1), puncta number steadily increases between early larval and L4 or adult stages. In the case of ASK, synapse addition – shown for all presynaptic specializations visualized with GFP::CLA-1 (*Figure 8B*) and for ASK>AIA synapses (*Figure 8C*), AIA being ASK's major post-synaptic partner (*White et al., 1986*) – occurs roughly in scale with the increase in neurite length (*Figure 8A*) in the nerve ring; specifically, there is a ~twofold increase across all three measures. In a previous study, a comparative analysis of the nerve ring from EM samples spanning multiple larval- and adult-staged *C. elegans* showed that system-wide neurite length increases with the length of the animal and that synapses are added progressively, thereby maintaining overall synapse density (*Witvliet et al., 2021*). Since this was done on a systemic scale, albeit with a small sample size, the study noted that synapse addition was likely a cell-specific feature with some neurons adding or pruning more synapses than others (*Witvliet et al., 2021*). Notably, 'hub' neurons added disproportionately more synapses across development. Similarly, a subset of neuron classes that undergo stage-specific plasticity (*Witvliet et al., 2021*) and/or synaptic pruning with the onset of sexual maturation (some examples are shown in *Figure 3*) remove synapses in a stage- and cell-specific manner.

Uniform synapse addition has also been noted in sparse EM-based comparative connectomic studies in other systems, such as the nociceptive circuits in *D. melanogaster* between first and third instar larval stages (*Gerhard et al., 2017*), motoneurons in *D. melanogaster* across larval development (*Couton et al., 2015*), antennal lobe in *D. melanogaster* between late pupa and adulthood (*Devaud et al., 2003*), and primary somatosensory and visual cortical regions in mice and rhesus macaque during early postnatal development (*Wildenberg et al., 2023*). Importantly, since our approach is high throughput, it allows us to ascertain synapse addition while accounting for inter-individual variability in synapse number in the reporters studied. This is a key point, which remains hard to address

with current EM output, but focusing on a 'core' set of synapses present across EM samples (*Cook et al., 2023*) presents an opportunity for statistically powered analyses.

## Quantitative analysis of synapses across development in the ENS

To broaden our analysis of developmental plasticity in *C. elegans*, we next focused on neurons in the pharynx, or ENS. The length of the pharynx from the tip of the corpus to the end of the terminal bulb increases twofold (from 60 to 130 µm, roughly) between L1 and adulthood (*Shibata et al., 2016*). This elongation is modest relative to the overall ~4.5-fold increase (from 250 to 1150 µm, roughly) in the length of the animal across the same developmental window (*Witvliet et al., 2021*; *Shibata et al., 2016*; *Figure 9A*). To investigate how neuronal processes and presynaptic specializations of pharyngeal neurons scale with the elongation of the whole organ, we quantified neurite length (denoted by 'Length of ROI') and the number of fluorescently tagged CLA-1 or RAB-3 puncta in the pharyngeal neuron classes MC, M3, I2, M4, and NSM (*Figure 9B–F*). In this case, puncta were counted both manually (*Figure 9*) and using WormPsyQi (*Figure 9—figure supplement 1*). Due to high synapse density and lack of or faint expression of a cytoplasmic reporter in most strains analyzed, WormPsyQi undercounted puncta in early larval stages (*Figure 9—figure supplement 1*). However, trends in developmental plasticity between manual and automated quantification were similar (*Figure 9—figure supplement 1*).

We found that in contrast to neurons in the CNS, neurons in the ENS show striking disparity in synapse addition across neuron classes. The number of CLA-1/RAB-3 puncta in neurons MC, I2, and M3 remains constant (neurons MC and I2, *Figure 9B, C*; *Figure 9—figure supplement 1A, D*) or increases modestly compared to the elongation of the pharynx (neuron M3, *Figure 9D*; *Figure 9—figure supplement 1B*) between early larval stages (L1 or L2) and adulthood. This is in spite of the neurite length scaling roughly in line with the length of the pharynx in all three cases (*Figure 9B–D*). In other words, synapse density in these neurons decreases across development. We note that in some neurons, such as M3 (*Figure 9D*; *Albertson and Thomson, 1976*), it is possible that the large variability in synapse number across individuals and within a bilateral pair, at a given stage in development, may complicate comparative analysis between stages. Furthermore, since our analysis does not include quantification of neuronal adjacency across development, we cannot rule out the possibility that physical contact within neighborhoods of pharyngeal neurons, which make synapses with one another is not scaling with the elongation of the pharynx or the whole animal. EM reconstructions, because of their often system-wide nature, may be better suited for this type of analysis since it yields quantification of both the connectome and the adjacencies, or 'contactome', between synaptic partners.

In contrast to neurons that show variable synapse density across developmental stages, we found two neuron classes, M4 and NSM, where CLA-1/RAB-3 puncta are added proportionately with increasing neurite length and overall pharynx elongation across development ('Total count' in *Figure 9E, F*; *Figure 9—figure supplement 1C*). Our CLA-1-based data for NSM also corroborate previous findings of the developmental trajectory of NSM's synaptic branches (*Axäng et al., 2008*). Based on previous EM studies (*Cook et al., 2020*; *Albertson and Thomson, 1976*), these puncta largely correspond to neuromuscular junctions, many of which arise post-embryonically and can be considered somewhat separate from the 'core' pharyngeal connectome established in the embryo which seems more static. To take a deeper look, we calculated M4 and NSM puncta number separately in the pharyngeal nerve ring and isthmus, and found synapse addition between L1 and adulthood in both regions (*Figure 9E, F*). Given the density of puncta in the nerve ring, it is ultimately difficult to discern changes in neuron-to-neuron synapses made by these neurons by relying solely on CLA-1/RAB-3 reporters. Synapse-specific reporters built using GRASP/iBLINC or temporal EM analysis will offer greater resolution needed to fully understand developmental plasticity in these synapse-dense regions. In conclusion, developmental synapse addition appears to differ between the ENS and CNS, as suggested by the lesser increase in synapse number in pharyngeal neurons relative to somatic neurons in our data (*Figures 8 and 9*).

## Using WormPsyQi to extract subcellular information of synapse distribution

Next, we exploited WormPsyQi to extract subcellular information of synapse distribution. To this end, we quantified spatial positions of presynaptic specializations in neurons PHA, AIB, and ASK as a proof of principle (see Methods). We found that the presynaptic reporter data recapitulated expected subcellular distribution patterns in all three neurons analyzed. Previous EM data (*Cook et al., 2019*; *White et al., 1986*) show that in PHA neurons, presynaptic sites are concentrated – and uniformly distributed – in the axonal region anterior to the branch point with close to no synapse posterior to it; our PHA-specific RAB-3 reporter recapitulates this topological distribution (*Figure 10A*). Taking advantage of our sizeable dataset, we took the analysis a step further and measured the probability of finding a synapse at a particular location along the axon, thus quantitatively showing the distribution of stereotyped (high probability density) and variable or noisy (low probability density) sites; we found that distribution was indeed more or less uniform in the case of PHA in spite of inter-individual variability (*Figure 10A*, *Figure 10—figure supplement 1A*).

We performed a similar analysis on CLA-1-based reporters for neurons AIB and ASK (*Figure 10*). AIB's axons occupy two distinct neighborhoods of the nerve ring (*Moyle et al., 2021*), anterior and posterior, with a majority of presynaptic sites localizing in the anterior region (*Sengupta et al., 2021*). Both RAB-3 (*Sengupta et al., 2021*; *Figure 4—figure supplement 1D*) and CLA-1 (*Figures 4A and 10B*) reporters for AIB qualitatively show this distribution. Quantification of topological synapse distribution within the anterior axon in a representative population revealed that synapse density is sparse in the ventral regions, but increases gradually as the processes travel to the crossover point at the dorsal midline, with highest density proximal to it. We also observed widespread variability in puncta size that could not be correlated with spatial location (*Figure 10—figure supplement 1B*). Lastly, in the case of ASK neurons, EM data show (*White et al., 1986*) distinct regions along the axons where synapses localize. Our RAB-3 (*Figure 4B*) and CLA-1 (*Figures 4A and 10C*) recapitulate this distribution pattern in a representative population. We found that ASK axon regions differed markedly in absence or presence of a synapse as well as stereotyped vs. variable sites as denoted by the distinct peaks in the estimated probability density plot (*Figure 10C*).

Having earlier quantified the developmental plasticity of synapse number in ASK neurons between L1 and adulthood (*Figure 8*), we next asked how the stereotyped topological distribution and synapse ultrastructure features (such as synapse volume) change across development. The total volume of synapses scaled proportionately with synapse addition between L1 and adult animals (*Figure 10D*), and peaks in probability density were stereotyped as well (*Figure 10E*), suggesting that these ultrastructure and spatial quantifications followed the same extent of plasticity as the overall number of synapses did. In addition, we plotted the number of puncta against ultrastructure features including mean volume and mean intensity, and found that mean volume, in the ASK>AIA reporter studied in detail, clustered in an age-dependent manner; conversely, the synapse density vs. mean intensity feature space did not show any age-dependent clustering (*Figure 10—figure supplement 2*).

Although this analysis is limited to a few neurons, the observation that synapse distribution – for synapses that are present at birth – remains stereotyped is not unanticipated given that, aside from inter-individual variability, the overall architecture of the brain in terms of cell body positions, neuronal morphologies, and relative arrangement of processes within neighborhoods in *C. elegans* are largely maintained across development (*Witvliet et al., 2021*; *Moyle et al., 2021*; *Brittin et al., 2021*). An exception to this organizational stereotypy is a subset of neurons that undergo drastic changes across one or several of these spatial features between L1 and adulthood, such as birth of new neurons (e.g. VNC neuron classes), migration of cell bodies (e.g. Q blast cell progeny), dendritic growth (e.g. FLP, PVD, and NSM), synapse pruning (e.g. sexually dimorphic neurons PHB, AVA, etc.), and neighborhood change (e.g. DD and VD neurons) (*Androwski et al., 2020*; *Emmons, 2018*; *Howell et al., 2015*; *Middelkoop and Korswagen, 2014*; *Altun and Hall, 2011*; *Oren-Suissa et al., 2010*; *Smith et al., 2010*; *Axäng et al., 2008*). Overall, precise quantification of synapse distribution patterns and ultrastructural features on a population level, something that can only be done qualitatively by eye, is a useful way of assigning topological information to reporter-based datasets and can provide valuable readouts under various genetic and environmental perturbations, especially in cases where the synapse number may remain unchanged.

## Limitations of the toolkits described

We have demonstrated that the toolkits described here enable a high throughput, unbiased, in-depth, and robust workflow for studying chemical and electrical synapses in *C. elegans* based on analyzing many reporters. However, we note some important limitations. First, the reporters presented here only mark single components of synapses. For instance, the NLG-1-based GRASP reagents used here specifically visualize NLG-1 localization and hence the interpretation of puncta number and features quantified with WormPsyQi only represent NLG-1 features and dynamics. Hence, this approach may not be ideal for neurons which do not endogenously express NLG-1 at the right time or localize it to synaptic domains, therefore under- or mis-representing the synapse population between two neurons.

Another limitation of the reporters described is that they contain over-expressed multicopy transgenic arrays, which could lead to trafficking and localization artifacts. While some of these problems are mitigated by injecting transgenes at different concentrations, integrating extrachromosomal arrays, and testing multiple gene promoters to drive expression, these measures may fail and may introduce a selection bias. In the case of transsynaptic technologies, where multiple transgenes need to be co-injected, artifacts tend to be harder to identify and eliminate. Comparisons of multiple reporters expressing an identical transgene often show variability across lines picked from the same injection. This holds true for both presynaptic and transsynaptic reporters, although the latter tend to have more variability across lines. This could partially explain discrepancies between data from transgenic reporters and EM reconstructions (*Figures 5, 6, 8*). With gene editing becoming more accessible, an increasing collection of reporters with endogenously tagged proteins known to localize in specific neurons and at specific sites along a neurite will add to our knowledge of pre- and post-synaptically localized proteins available to visualize synapses. Reporters with endogenously tagged proteins and single-copy transgenes show more consistent expression and spatial localization. Indeed, the use of diverse types of reporters, when available, will be essential to establish the ground-truth and inform mutant analyses for advancing our understanding of synapse biology.

Lastly, although we have shown that the image analysis pipeline is robust and versatile, it is constrained by the type of images and reporters being analyzed. In cases where the cytoplasmic marker is suboptimal (e.g. dim cytoplasmic marker in CLA-1 reporters for pharyngeal neurons I2 and NSM, or dim signal on the far end of the coverslip while acquiring images), the resulting neurite mask may be discontinuous, resulting in undercounting of synaptic puncta in downstream steps. In such scenarios, it may be useful to visualize the neurite mask in Fiji prior to synapse quantification. Since this technical constraint is independent of WormPsyQi but can directly affect its performance, we suggest resolving this prior to processing an entire dataset. For instance, one can use a stronger promoter to drive the cytoplasmic fluorophore, perform some preprocessing such as deconvolution, denoising, or contrast enhancement prior to synapse segmentation, or skip the masking step altogether and directly segment synapses. The last scenario is particularly useful in cases where the synapse signal is brighter than the cytoplasmic reporter; for our quantification, we have relied on this for several reporters and denoted them with 'no mask' in *Supplementary file 1*. We note that the processing time is significantly reduced in the presence of a mask, so if the user decides to skip masking prior to synapse segmentation, we suggest cropping the precise ROI so the quantification can be expedited. Also, where the synapse signal is diffuse (e.g. RAB-3 reporters) or synapse density is too high (e.g. CLA-1/RAB-3 reporters for IL2, NSM, AIB, etc., *Figures 4 and 6*), quantification can be inaccurate. Analyzing the 'overlay' images saved during image processing can reveal problematic regions of interest and help the user decide whether to proceed with quantification. In general, if the fluorescent signal in an image is visibly discrete, WormPsyQi works well in quantifying many discernible and subtle features that would otherwise be tedious or impossible to achieve.

# Materials and methods

## Strains and maintenance

Wild-type strains used were *C. elegans* Bristol (strain N2). Worms were grown at 20°C on nematode growth media plates seeded with bacteria (*E. coli* OP50). Details of all transgenic strains are listed in *Supplementary file 1*; corresponding sex and age are also noted.

## Cloning and constructs

All constructs used to build the synaptic reporters are listed in *Supplementary file 2*. In most cases, subcloning by restriction digestion was performed to generate GRASP, CLA-1, and cytoplasmic (TagRFP and mCherry) constructs. Cell-specific promoter fragments were amplified from N2 genomic DNA and cloned into 5′ SphI- and 3′ XmaI-digested vectors expressing fluorophore only (in the case of cytoplasmic constructs), GFP::CLA-1, GFP::RAB-3, and split-GFP (spGFP1-10 and spGFP11) sequences; T4 ligation was used to insert digested promoter fragments into digested vectors. A plasmid containing GFP::CLA-1(S) (PK065) was kindly provided by Peri Kurshan. In some cases, RF-cloning or Gibson Assembly (NEBuilder HiFi DNA Assembly Master Mix, Catalog # E2621L) were used as the preferred choice of cloning. Primers for specific promoter fragments are listed in *Supplementary file 2*.

Transgenic strains were generated by microinjecting constructs as simple extrachromosomal arrays into N2 or *him-5 (e1490)* genetic backgrounds. Precise concentrations of co-injected constructs are detailed in *Supplementary file 2*. Extrachromosomal array lines were selected according to standard protocol. Where integration was performed, gamma irradiation was used.

## Microscopy and image analysis

Worms were anesthetized using 100 mM sodium azide (NaN$_3$) and mounted on 5% agar on glass slides. Worms were analyzed by Nomarski optics and fluorescence microscopy, using a ×40 or ×63 objective on a confocal laser-scanning microscope (Zeiss LSM880 and LSM980) or a spinning disk confocal (Nikon W1). When using GFP, we estimated the resolution of our confocal to be ~250 nm. 3D image Z-stacks were converted to maximum intensity projections using Zeiss Zen Blue or ImageJ software (*Schindelin et al., 2012*). Manual quantification of puncta was performed by scanning the original full Z-stack for distinct dots in the area where the processes of the two neurons overlap. Figures were prepared using Adobe Illustrator.

## Statistical analysis

Statistical tests were performed using Python (version 3.10) or GraphPad PRISM (version 9.5.1). When multiple tests were performed, *post hoc* Bonferroni correction was used to adjust p values for the number of pairwise comparisons made.

## Neurite segmentation

### Model structure

We adapted a '2.5D' U-Net model (*Guo et al., 2020*) to segment neurites from 3D fluorescent image stacks. The input is a 3D chunk with dimension $7 \times 256 \times 256$ ($D \times H \times W$), and the prediction is of the center slice ($256 \times 256$). The chunk that includes nearby slices provides more spatial context compared to 2D segmentation, while reducing the memory constraints of 3D segmentation. The model has an encoding block and a decoding block with skip connections at the same resolution level that 'squeeze' 3D into 2D (*Figure 2—figure supplement 2A*). The structure of the encoding block is detailed in *Figure 2—figure supplement 2A'*.

### Training process

The U-Net model was trained using a 3D fluorescent microscopy dataset. The dataset covers 5 neurons from different developmental stages (ASK, AIA, PHB, AVA, and I5), with 16 images in total. We split the dataset into the ratio 10:2:4 for training, validation, and testing, respectively. Due to the extremely large size of the original image stack, we sampled each 3D stack randomly into 200 smaller 3D patches as stated above and filtered out the blank background patches using a threshold on mean intensity. For the training process, we used a combination of IoU loss (*Berman et al., 2017*) and the cross entropy loss: $L = L_{IoU} + L_{CE}$, with the learning rate set to 0.001. The training usually converges within 200 epochs. Data augmentation (including random flipping and rotation) is added for each epoch. The detailed benchmark summary is shown in *Supplementary file 3*. All training experiments were done using the same dataset with fourfold cross-validation, and same loss function and batch size. The metrics shown in the table refer to the neurite class, except for accuracy, which shows an overall pixel prediction accuracy. Training and inference were done on a Nvidia V100 16 GB GPU.

## Synapse segmentation

### Process overview

To segment synapses from 3D fluorescent image stacks, WormPsyQi uses a two-layer pixel classification model (*Figure 2—figure supplement 2B*). In the first layer, a set of predefined 2D and 3D local filters are applied to each pixel to generate a feature vector. The first-layer classifier then estimates the probability that a pixel belongs to a synapse based on this feature vector. In the second layer, the probability estimations of a pixel and its nearby pixels are added to the first layer's feature vector to create an extended vector. This extended vector is then fed into the second-layer classifier to determine whether the pixel belongs to a synapse. Optionally, the binary neurite mask from the neuron segmentation step is used to filter out the false-positive pixels that are not in the vicinity of synapse-forming regions. Although we primarily used the Random Forest (RF) for the classifiers in this paper, WormPsyQi provides RF, Support Vector Machine (SVM), and multilayer perceptron as options for training a new model with the GUI. While SVM is effective in many different machine learning applications and generally less prone to overfitting with a small dataset, it is sensitive to noise and its fit time complexity is more than quadratic, making it less ideal for large training datasets. During training, grid search hyperparameter optimization with fivefold cross-validation is used. For prediction, if the target image has a neurite mask segmented in the previous step, only the pixels in the mask are sampled to be predicted by the trained synapse segmentation model; otherwise, all pixels are subject to prediction. The local and probability features are extracted similarly to the training step, and pixels outside of the neurite mask are automatically classified as background.

### Training data

One or more pairs of a fluorescent microscopy image and its corresponding binary label image are required to train the model (*Figure 2*). For the pre-trained models and strain-specific models used in this work, the pencil tool in Fiji (*Schindelin et al., 2012*) was used to generate binary label images.

### Training of the first layer

The first-layer classifier is trained through two sequential training steps (*Figure 2—figure supplement 2B*). In the first step, all positive pixels and a small portion of negative pixels near the positive ones are selected as a training set. Local features of these pixels are extracted and fed along with the label to train the first-layer classifier. Since only a small portion of negative pixels spatially close to the positive pixels is included in the initial training dataset, the first-layer classifier is particularly prone to type I error after this step. To resolve this issue, more negative pixels are added to the training dataset in the second step, and the classifier is retrained. After the first step, pixels in sampled patches are predicted by the trained classifier and false negative pixels are selected to be added to the training dataset. These target patches are sampled in a similar manner as before but are ten times larger in dimension. The first-layer classifier is then fitted again with this expanded training dataset.

### Training of the second layer

As the size of a synapse is generally bigger than just a single pixel, a pixel adjacent to the positive foreground is more likely to be foreground than another pixel with the same local features but away from the foreground. To account for this factor, we added the probabilities of adjacent pixels being positive as additional features to the original local features for the second-layer classifier. For each pixel in the training dataset, probabilities of the target pixel and the six adjacent pixels ($-1$ and $+1$ in either $x$, $y$, or $z$ direction) are calculated by the first-layer classifier and the feature vector is expanded with these six probability values. The second-layer classifier is then trained with the expanded features using the same model as the first-layer classifier (*Figure 2—figure supplement 2B*).

### Local features

The local features used in WormPsyQi's synapse classifier include the pixel value, Sobel derivatives, difference of Gaussians, Laplacian of Gaussian, local average and standard deviation, and erosion of both synapse and neurite channels. Before training, each feature is normalized with a standard scaler, removing the mean and scaling to unit variance.

## Patch sampling in training

Among the pixels in the label images, only a small portion of pixels in the patches surrounding positive labels is selected to be fed into the model as training data points (*Figure 2—figure supplement 2B*). This step not only reduces the time and memory required for the training process but also results in higher accuracy by preventing the model from fitting to irrelevant data points. Each 2D slice in the original microscopy image is first divided into non-overlapping patches, and only the patches containing positive pixels in the corresponding patch in the label image are processed in the following steps. The size of the patches is set to twice the square root of the average area of the connected labels.

## Pre-trained synapse segmentation models

WormPsyQi provides four different pre-trained synapse segmentation models: (1) labeled as 'GRASP (sparse)' in the GUI, trained with three *otIs612* (PHB>AVA GRASP) images, (2) 'GRASP (dense)', trained with three *otIs653* (ASK>AIA GRASP) images, (3) 'CLA-1', trained with six *otEx7503* (I5 CLA-1) images, and (4) 'RAB-3', trained with two *otEx7231* (ASK RAB-3). RF model was used for all of them.

## Synapse distribution quantification

The relative locations of synaptic puncta along neurites were determined using synapse segmentation masks and neurite skeletons. The skeletons (*Figure 10*) were extracted on top of neurite segmentation masks generated by the pipeline, coupled with minor manual editing using NeuTube (*Feng et al., 2015*) in cases where the segmentation result was not perfect. For each synapse, its location was marked by the closest node of the skeleton with respect to its centroid coordinates. When a synapse was too large and unresolved, each pixel of the synapse was used instead of the centroid to give a smoother distribution. The distribution was normalized from 0 to 1 either by the entire neurite length (from soma to the axon terminal), or by synapse domain length (neurite segment occupied by synapses). The probability density of distribution of a population was acquired by combining all worms of the population and dividing the entire synaptic volume.

## Data and code availability

The WormPsyQi software is implemented with Python and is publicly available at https://github.com/lu-lab/worm-psyqi (*Han, 2023*). PyTorch (*Paszke et al., 2019*) is used for implementing the deep CNN model in synapse segmentation process, and Scikit-learn (*Pedregosa et al., 2012*) package is mainly used for other methods. A comprehensive guide to installing and operating the software is on the Github repository. To facilitate use, the code is accompanied by a user manual with instructions for synapse quantification that is both standardized to yield quantitative data and flexible for customization. All analyzed datasets are uploaded in a Zenodo repository, which can be accessed here: https://zenodo.org/records/10368858.

## Acknowledgements

We thank Chi Chen for assistance with microinjections to generate strains; Ulkar Aghayeva for generating RAB-3 reporters for pharyngeal neurons MC, M3, and M4; Steven Cook, Emily Bayer, and Cyril Cros for kindly sharing previous image datasets for synapse analysis. This work was funded by an NSF NeuroNex award (#1707401) and NSF-Simons Southeast Center for Mathematics and Biology (#1764406).

# Additional information

## Funding

| Funder | Grant reference number | Author |
|---|---|---|
| National Science Foundation | 1707401 | Oliver Hobert<br>Hang Lu |
| National Science Foundation Simons Southeast Center for Mathematics and Biology | 1764406 | Hang Lu |

The funders had no role in study design, data collection, and interpretation, or the decision to submit the work for publication.

## Author contributions

Maryam Majeed, Conceptualization, Data curation, Formal analysis, Investigation, Writing – original draft, Writing – review and editing; Haejun Han, Keren Zhang, Conceptualization, Data curation, Formal analysis, Investigation, Writing – review and editing; Wen Xi Cao, Chien-Po Liao, Formal analysis, Investigation, Visualization, Writing – review and editing; Oliver Hobert, Hang Lu, Conceptualization, Funding acquisition, Project administration, Writing – review and editing

## Author ORCIDs

Maryam Majeed ⓘ http://orcid.org/0000-0003-4018-0766
Haejun Han ⓘ http://orcid.org/0000-0003-1822-8567
Oliver Hobert ⓘ http://orcid.org/0000-0002-7634-2854
Hang Lu ⓘ http://orcid.org/0000-0002-6881-660X

Reviewer #1 (Public Review): https://doi.org/10.7554/eLife.91775.3.sa1
Reviewer #2 (Public Review): https://doi.org/10.7554/eLife.91775.3.sa2
Reviewer #3 (Public Review): https://doi.org/10.7554/eLife.91775.3.sa3
Author Response https://doi.org/10.7554/eLife.91775.3.sa4

# Additional files

## Supplementary files

• Supplementary file 1. List of fluorescent reporters analyzed in this paper using WormPsyQi. Strain identifier, genotype, datasets analyzed, and imaging conditions (stage, sex) are listed for each reporter.

• Supplementary file 2. Details of transgenic constructs and injections.

• Supplementary file 3. Segmentation benchmark for different U-Net structures.

• MDAR checklist

## Data availability

All data generated or analyzed during this study are included in the manuscript and supporting file. Strains are available at the CGC. Code is available at Github, as detailed in the manuscript. Primary data are uploaded at Zenodo, as indicated in the manuscript (https://zenodo.org/records/10368858).

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
