## [Editor Report · eLife assessment]

Studies of synaptic development and plasticity in the nematode *C. elegans* have been limited by the difficulty of rapid, accurate assessments of synaptic structure. Here, with a series of **convincing** studies, the authors introduce and validate a **valuable** computational pipeline, "WormPsyQi," that allows rapid, reproducible quantitation of fluorescent synaptic puncta while minimizing human error and bias. The authors also describe a new set of strains carrying synaptic markers. Together, these tools should provide groups studying this model system with the ability to quantitatively characterize chemical and electrical synapses, even in densely packed regions in 3D space such as the nerve ring.

---

## [Referee Report · Reviewer #1 (Public Review)]

Summary:

The paper by Majeed et al has a valuable and worthwhile aim: to provide a set of tools to standardize the quantification of synapses using fluorescent markers in the nematode *C. elegans*. Using current approaches, the identification of synapses using fluorescent markers is tedious and subject to significant inter-experimenter variability. Majeed et al successfully develop and validate a computational pipeline called "WormPsyQi" that overcomes some of these obstacles and will be a powerful resource for many *C. elegans* neurobiologists.

Strengths:

The computational pipeline is rigorously validated and shown to accurately quantitate fluorescent puncta, at least as well as human experimenters. The inclusion of a mask - a region of interest defined by a cytoplasmic marker - is a powerful and useful approach. Users can take advantage of one of four pre-trained neural networks, or train their own. The software is freely available and appears to be user-friendly. A series of rigorous experiments demonstrates the utility of the pipeline for measuring differences in the number of synaptic puncta between sexes and across developmental stages. Neuron-to-neuron heterogeneity in patterns of synaptic growth during development are convincingly demonstrated. Weaknesses and caveats are realistically discussed.

---

## [Referee Report · Reviewer #2 (Public Review)]

Summary:

This paper nicely introduces WormPsyQi, an imaging analysis pipeline that effectively quantifies synaptically localized fluorescent signals in *C. elegans* through high-throughput automation. This toolkit is particularly valuable for the analysis of densely packed regions in 3D space, such as the nerve ring. The authors applied WormPsyQi to various aspects, including the examination of sexually dimorphic synaptic connectivity, presynaptic markers in eight head neurons, five GRASP reporters, electrical synapses, the enteric nervous system, and developmental synapse comparisons. Furthermore, they validated WormPsyQi's accuracy by comparing its results to manual analysis.

Strengths:

Overall, the experiments are well done, and their toolkit demonstrates significant potential and offers a valuable resource to the *C. elegans* community. This will expand the range of possibilities for studying synapses in *C. elegans*.

---

## [Referee Report · Reviewer #3 (Public Review)]

Summary:

In this manuscript, the authors present a new automated image analysis pipeline named WormPsyQi which allows researchers to quantify various parameters of synapses in *C. elegans*. Using a collection of newly generated transgenic strains in which synaptic proteins are tagged with fluorescent proteins, the authors showed that WormPsyQi can reliably detect puncta of synaptic proteins, and measure several parameters, including puncta number, location, and size.

Strengths:

The image analysis of fluorescently labeled synaptic (or other types of) puncta pattern requires extensive experience such that one can tell which puncta likely represent bona fide synapse or background noise. The authors showed that WormPsyQi nicely reproduced the quantifications done manually for most of the marker strains they tested. Many researchers conducting such types of quantifications would receive significant benefits in saving their time by utilizing the pipeline developed by the authors. The collection of new markers would also help researchers examine synapse patterning in different neuron types which may have unique mechanisms in synapse assembly and specificity. The authors describe the limitations of the use of toolkits and potential solutions users can take.

---

## [Author Response]

The following is the authors’ response to the original reviews.

**Reviewer #1 (Recommendations For The Authors):**
I have only a few very minor suggestions for improvement.the text repeatedly uses the terms "central nervous system" and "enteric nervous system", which are not in standard use in the field. These terms are not defined until the bottom of p. 12 even though they are used earlier. It would be useful for the authors to explicitly describe their definitions of these terms earlier in the paper.

Fixed.

the inclusion of four pre-trained models is a powerful and useful aspect of WormPsyQi. Would it be possible to develop a simple tool that, when given the user's images, could recommend which of the four models would be most appropriate?

We appreciate the reviewer for bringing this up. To address this, we have now added an additional function in the pipeline to test all pre-trained models on representative input images. Before processing an entire dataset, users can view all segmentation results for images in Fiji to assess which model performed best, judged by the user. The GUI, running guide document, and manuscript have been modified accordingly.

In addition, we would like to emphasize that the pre-trained models were developed by iterative analyses of many reporters, often with multiple rounds of parameter tuning; the results were validated post hoc to choose the optimal model for each reporter, and we have listed this information in Supplemental Table 1 to inform the choice of the pre-trained model for commonly used reporter types.

On p. 11 (and elsewhere), the differences in the performance of WormPsyQi and human experimenters are called "statistically insignificant". This statement is not particularly informative (absence of evidence is not evidence of absence). Can the authors provide a more rigorous analysis here - or provide an estimate of the typical effect size of the machine-vs-human difference?

To address this, we have included additional analysis in Figure 2 – figure supplement 3. For two reporters - I5 GFP::CLA-1 and M4 GFP::RAB-3 - we compare WormPsyQi vs. labelers and inter-labeler puncta quantification. A high Pearson correlation coefficient (r2) reflects greater correspondence between two independent scoring methods. We chose these two test cases to demonstrate that the machine-vs-human effect size is reporter-dependent. For I5, where the CLA-1 signal is very discrete and S/N ratio is high, the discrepancy between WormPsyQi, labeler 1, and labeler 2 is minimal (r2=0.735); moreover, scoring correspondence depends on the labeler (r2=0.642 and 0.942, respectively). In other words, WormPsyQi mimics some labelers better than others, which is to be expected. For M4, where the RAB-3 signal is diffuse and synapse density is high in the ROI, the inter-labeler discrepancy is high (r2=0.083) and WormPsyQi vs labeler (1 or 2) discrepancy is slightly reduced (r2=0.322 and 0.116, respectively). The problematic regions for the M4 RAB-3 reporter are emphasized in Figure 6 - figure supplement 1A. Overall, the additional analysis suggests that the effect size is contingent on the reporter type and image quality, and importantly for scoring difficult strains WormPsyQi may average out inter-labeler scoring variability.

p. 12: "Again, relying on alternative reporters where possible..." This is an incomplete sentence - are some words missing?

Edited.

**Reviewer #2 (Recommendations For The Authors):**
1. The authors effectively validated the sexually dimorphic synaptic connectivity by comparing the synapse puncta numbers of PHB>AVA, PHA>AVG, PHB>AVG, and ADL>AVA. However, these differences appear to be quite robust. It would be beneficial for the authors to test whether WormPsyQi can detect more subtle changes at the synapses, such as 10-20% changes in puncta number and fluorescence intensity.

While the dimorphic strains were used to first validate WormPsyQi based on the ground truth of very well-characterized reporters, the reviewer reasonably asks whether our pipeline can pick up on more subtle differences. To address this, we have now included an additional figure (Figure 9 – figure supplement 2), where we performed pairwise comparisons between L4 and adult timepoints for the reporter M3 GFP::RAB-3. As reflected in panels A and C, although the difference between puncta number and mean intensity between L4 and adult is marginal (22% increase in puncta number and 13% increase in mean intensity from L4 to adult), WormPsyQi can pick it up as statistically significant.

2. On page 10, the authors mentioned that "cell-specific RAB-3 reporters have a more diffuse synaptic signal compared to the punctate signal in CLA-1 reporters for the same neuron, as shown for the neuron pair ASK (Figure 4 -figure supplement 1B, C)". It is important to note that in this case, the reporter gene expressing RAB-3 is part of an extrachromosomal array, whereas the reporter gene expressing CLA-1 is integrated into the chromosome. It's possible that the observed difference in pattern may arise from variations in the transgenic strategies employed.

To emphasize the difference in puncta features inherent to the reporter type, we have now added WormPsyQi segmentation results for ASK CLA-1 extrachromosomal reporter (otEx7455) next to the ASK CLA-1 integrant (otIs789) and ASK RAB-3 reporter (otEx7231) in Figure 4 – figure supplement 1C. Importantly, otEx7455 was integrated to generate otIs789, so they belong to the same transgenic line. Literature shows that RAB-3 and CLA-1 have different localization patterns and corresponding functions at presynaptic specializations, and this is qualitatively and quantitatively shown by the significant difference in puncta area size between RAB-3 and both CLA-1 reporters, i.e., both CLA-1 reporters have smaller, discrete puncta compared to RAB-3 (Figure 4 – figure supplement 1C). Quantitatively, in the case of ASK - where the synapse density is sparse enough that even diffuse RAB-3 puncta can be segmented without confounding adjacent puncta – overall puncta number between otEx7231 and otIs789 are similar. However, RAB-3 signal is diffuse and this poses quantification problems in cases where the synapse density is higher (e.g. AIB, SAA in Figure 4 – figure supplement 1D) and WormPsyQi fails to score puncta in these reporters since the signal is not punctate. As far as integrated vs. extrachromosomal reporters go, the reviewer is right in pointing out that some differences may be stemming from reporter type as our additional analysis between otIs789 and otEx7455 indeed shows fewer puncta in the latter owing to variable expressivity.

3. The authors mentioned that having a cytoplasmic reporter in the background of the synaptic reporter enhanced performance. It would be more informative to provide comparative results with and without cytoplasmic reporters, particularly for scenarios involving dim signals or densely distributed signals.

The presence of a cytoplasmic marker is critical in two specific scenarios: (1) images where the S/N ratio is poor, and (2) when the image S/N ratio is good, but the ROI is large, which would make the image processing computationally expensive.

To demonstrate the first scenario, we have included an additional panel in Figure 4 – figure supplement 1(B) to show how WormPsyQi performs on the PHB>AVA GRASP reporter with and without the channel having cytoplasmic marker. The original image was processed as-is in the former case with both the synaptic marker in green and cytoplasmic marker in red; for comparison, only the green channel having synaptic marker was used to simulate a situation where the strain does not have a cytoplasmic marker. As shown in the figure, in the presence of background autofluorescence signal from the gut (which can be easily confounded with GRASP puncta depending on the worm’s orientation), WormPsyQi quantified GRASP puncta much more robustly with the cytoplasmic label; without the cytoplasmic marker, gut puncta are incorrectly segmented as synapses (highlighted with red arrows) while some dim synaptic puncta are not picked up (highlighted with yellow arrows).

To demonstrate the second scenario, we now highlight the case of ASK CLA-1 in Figure 2 - figure supplement 4E. Additionally, we have emphasized in the manuscript that in cases where the S/N ratio is good and the image is restricted to a small ROI, WormPsyQi will perform well even in the absence of a cytoplasmic marker. This is equally important to note as having a specific cytoplasmic marker in the background may not always be feasible and, in fact, if the cytoplasmic marker is discontinuous or dim relative to puncta signal, using a suboptimal neurite mask for synapse segmentation would result in undercounting synapses.

4. On page 12, the author stated "We also note that in several cases, GRASP quantification differed from EM scoring". However, the EM scoring is primarily based on a single sample, making it challenging to conduct a statistical analysis for the purpose of comparison.

This is correct and is indeed a limitation of EM for this type of analysis. We have now reworded this sentence (page 14) to emphasize the reviewer’s point, and it is also elaborated further in the limitations section.

5. In Figure 6F, the discrepancy between WormPsyQi and human quantification in the analysis of RAB-3 is observed. The author stated that "the RAB-3 signal was too diffuse to resolve all puncta". To better illustrate this discrepancy, it would be beneficial to include images highlighting the puncta that WormPsyQi cannot score, providing direct evidence that diffusing signals are not able to automatically detectable.

To highlight puncta that were not segmented by WormPsyQi but were successfully scored manually, we have included arrows in Figure 6. In addition, for reporter M4p::GFP::RAB-3, we have included magnified insets in Figure 6 - figure supplement 1A to highlight the region where human annotator scores more puncta than WormPsyQi owing to the high synapse density. In future implementations, additional functionality can be built for separating these merged puncta into instances based on geometrical features such as shape and intensity contour.

6. In Figure 9 S1D, the results from WormPsyQi and the manual are totally different. To address this notable discrepancy, the authors should highlight and illustrate the areas of discrepancy in the images. This visual representation can assist future users in identifying signal types that may not be well-suited for WormPsyQi analysis and inspire the development of new strategies to tackle such challenges.

This is now addressed in additional figure panels in Figure 4 – figure supplement 1B and Figure 6 - figure supplement 1A.

**Reviewer #3 (Recommendations For The Authors):**
I found the comparison between manual quantification and WormPsyQi-based quantification to be very informative. In my opinion, quantifying the number of puncta is not the most tedious/difficult quantification even when done manually. Would the authors be able to include manual-WormPsyQi comparison for more time-consuming and potentially more prone to human error/bias quantifications such as puncta size or distribution patterns using a few markers with some inter/intra animal variabilities?

To address this point, we have now included an additional figure supplement to Figure 2 (Figure 2 – figure supplement 4). We focused on the ASK GFP::CLA-1 reporter and had two human annotators manually label the masks of puncta for each worm by scanning Z-stacks and drawing all pixels belonging to each puncta in Fiji, which were then processed by WormPsyQi’s quantification pipeline to score puncta number, volume, and distribution. We also included a comparison of overall image processing time for each annotator and WormPsyQi. For features analyzed, the difference between WormPsyQi and human annotators for ASK CLA-1 is not statistically significant for multiple puncta features. Importantly, WormPsyQi reduces overall processing time by at least an order of magnitude, and while this is already advantageous for counting puncta, it is especially useful for other important puncta features since (a) they may not be easily discernible, and (b) it is extremely laborious to quantify them manually in large datasets when pixel-wise labels are required.

The authors listed minimum human errors and biases as one of the benefits of WormPsyQi. For the markers with discrepancies in quantifications between human and WormPsyQi, have the authors encountered or considered human errors/biases as potential reasons for such discrepancies?

This is the same point brought up by reviewer 1. We added Figure 2- figure supplement 3 to compare WormPsyQi to different human labelers, and show that because human labels can introduce systematic bias, WormPsyQi reduces such bias by scoring images using the same metric.

The authors noted that WormPsyQi would be useful for comparing different genotypes/environments. Some mutants have known changes in synapse patterning/number. It would be helpful if the authors could validate WormPsyQi using some of the mutants with known synapse defects. For instance, zig-10 mutant increases the cholinergic synapse density just by a bit (Cherra and Jin, Neuron 2016), and nlr-1 mutant disrupts punctated localization of UNC-9 gap junction in the nerve ring (Meng and Yan, Neuron 2020), which could only be detectable by experts' eyes. It would be interesting to see if WormPsyQi picks up such subtle phenotypes.

We agree that our pipeline would need to be tested in multiple paradigms to test its performance on detecting additional subtle phenotypes. In the context of this paper, we note that the developmental analysis of puncta in Figure 8 was performed to validate the ground truth from previous EM-based analyses (Witvliet et al., 2021), albeit the latter was limited by sample size. We extended this developmental analysis to the pharyngeal reporters, and in some cases the difference across timepoints was marginal (as emphasized by additional Figure 9 - figure supplement 2), but still detected by WormPsyQi. Lastly, our synapse localization analysis in Figure 10 assigns the probability of finding a synapse at a particular location along a neurite, which is not easily discernible by manual scoring.

One of the benefits of the automated data analysis program is to be able to notice the differences you do not expect. For example, there are situations where you feel that in certain genotypes there is something different from wild type with their synapses but you can't tell what's different from wild type. In such cases, you may not know what to quantify. I think it would be beneficial if there were more parameters to be included in the default qualifications such as puncta number/size/intensity/distributions in the pipeline, so that the users may find unexpected phenotypes from one of the default quantifications.

We apologize if this was not clearer in the manuscript where we first describe the pipeline in detail. To clarify, the output of WormPsyQi is a CSV file which includes several quantitative features, such as mean/max/min fluorescence intensity, puncta volume, and position. While most of our analyses are focused on puncta count, the user can perform downstream statistical analyses on all additional features scored to infer which features are most significantly variable across conditions. To make this clearer, we have elaborated the text when we first describe our pipeline, and along with the new Figure 2 - figure supplement 4, we hope that this point is clearer now.

In addition, most proof-of-principle analysis we performed was focused on an ROI where we expect the synapses to localize. In practice, the user can input images and perform quantification across the entire image without biasing toward an ROI (this can be done in the GUI synapse corrector window) to also evaluate synaptic changes in regions outside the usual ROI.

The authors stated that WormPsyQi could mitigate the problems stemming from scoring images with low signal-to-noise ratio or in regions with high background autofluorescence, laboriousness of scoring large datasets, and inter-dataset variability. Other than the 'laboriousness of scoring large datasets' it appeared to me that WormPsyQi does not do better than manual quantifications, especially inter-dataset variability, as the authors noted variability among the transgenes as one of the limitations of the toolkits. If two datasets are taken with completely different setups such as two independent arrays taken with two distinct confocal microscopes, would WormPsyQi make these two datasets comparable?

We have included additional figure supplements to address the reviewer’s point. A significant advantage WormPsyQi offers over manual scoring is that it provides a standardized method of quantifying synapse features. As shown in Figure 2 – figure supplement 3, human labelers can introduce systematic bias (e.g. some over count puncta, while some undercount). In addition, while puncta number may be relatively easy to quantify, especially in a high-quality dataset, more subtle puncta features such as size, intensity, and distribution are much more laborious to quantify and require a priori knowledge of signal localization (Figure 2 – figure supplement 4, Figure 10). Altogether, our pipeline facilitates multiple measurements while also enabling robust quantification in hard-to-score cases such as the example shown for PHB>AVA reporter (Figure 4 - figure supplement 1B).

Minor comments:Limitations are not quite specific to this work but those are general limitations to the concatemeric trans genes and fluorescently labeled synaptic proteins. I'd appreciate discussing specific limitations to WormPsyQi related to image acquisitions. For instance, for neurons with 3D structures would WormPsyQi be able to handle z-stacks closer to coverslip and stacks that are deeper side in a similar manner? Would the users need to be aware of such limitations when comparing different genotypes?

To address the reviewer’s comment, we have elaborated the last paragraph in the limitations section to explicitly discuss where the user should exercise caution. The reviewer reasonably points out that the fluorescent signal away from the cover slip is typically dimmer, and neurite masking in this case is indeed compromised if dim to start with. In such cases, we recommend that the user either performs some preprocessing such as deconvolution, denoising, or contrast enhancement to boost the neurite signal, or segment synapses without the neurite mask if the puncta signal is brighter than that of the cytoplasmic marker. We hope that our additional figure supplements will clarify that WormPsyQi’s performance is contingent on reporter type and image quality, thus making it easier for the user to discern where automated quantification falls short and alternative reporters should be explored. In general, if puncta are not discernible to the user due to very poor S/N ratio, for instance, we do not recommend using WormPsyQi to process such datasets; this will be manifest in the results of the new “test all models” feature we added in the revised version.

Some Rab-3 fusion proteins are described as RAB-3::GFP(BFP). Do these represent the C-terminal fusion of the fluorescent proteins? RAB-3 is a small GTPase with a lipid modification site at its C-terminus essential for its localization and function. Is it possible that the diffuse signal of some RAB-3 markers is caused by c-terminal fusion of the fluorescent protein?

While we do have reporters with N- and C-terminal RAB-3 fusions for different neurons, we do not have both for the same neuron to perform a fair comparison. However, as noted in response to a previous comment by reviewer 2, RAB-3 and CLA-1 have distinct localization patterns at the synapse and this aligns with their distinct functions: while RAB-3 localizes at synaptic vesicles, CLA-1 is an active zone protein required for synaptic vesicle clustering. Accordingly, we have observed diffuse RAB-3 signal in reporters irrespective of where the protein is tagged, and while this is not problematic for ROIs with a low synapse density, it confounds quantification in synapse-dense regions. In contrast, CLA-1 puncta are typically easier to quantify more discretely, which is particularly relevant for features such synapse distribution, size, and intensity.